# Mixture-of-Experts Operator Transformer for Large-Scale PDE Pre-Training

**Hong Wang**[1,2,3*], **Haiyang Xin**[1*], **Jie Wang**[1,2,3†], **Xuanze Yang**[1], **Fei Zha**[1],
**Huanshuo Dong**[1,2,3], **Yan Jiang**[1†]

[1] University of Science and Technology of China
[2] CAS Key Laboratory of Technology in GIPAS, University of Science and Technology of China
[3] MoE Key Laboratory of Brain-inspired Intelligent Perception and Cognition, University of Science and Technology of China

wanghong1700@mail.ustc.edu.cn, xhy2878@mail.ustc.edu.cn, jiewangx@ustc.edu.cn

## Abstract

Pre-training has proven effective in addressing data scarcity and performance limitations in solving PDE problems with neural operators. However, challenges remain due to the heterogeneity of PDE datasets in equation types, which leads to high errors in mixed training. Additionally, dense pre-training models that scale parameters by increasing network width or depth incur significant inference costs. To tackle these challenges, we propose a novel **M**ixture-**o**f-**E**xperts **P**re-training **O**perator **T**ransformer (**MoE-POT**), a sparse-activated architecture that scales parameters efficiently while controlling inference costs. Specifically, our model adopts a layer-wise router-gating network to dynamically select 4 routed experts from 16 expert networks during inference, enabling the model to focus on equation-specific features. Meanwhile, we also integrate 2 shared experts, aiming to capture common properties of PDE and reduce redundancy among routed experts. The final output is computed as the weighted average of the results from all activated experts. We pre-train models with parameters from 30M to 0.5B on 6 public PDE datasets. Our model with 90M activated parameters achieves up to a 40% reduction in zero-shot error compared with existing models with 120M activated parameters. Additionally, we conduct interpretability analysis, showing that dataset types can be inferred from router-gating network decisions, which validates the rationality and effectiveness of the MoE architecture [1].

## 1 Introduction

Learning solution operators for partial differential equations (PDEs) has emerged as a fundamental paradigm in scientific machine learning, enabling data-driven modeling of complex physical systems through neural operators [62, 22, 28, 12]. These operators learn mappings between infinite-dimensional function spaces, offering surrogate models that can outperform traditional numerical solvers by orders of magnitude in speed [45, 2]. To address the scarcity of PDE data and further enhance the performance of neural operators, recent studies have introduced pre-training techniques into neural operator frameworks [15]. However, their application to PDE learning remains nascent due to unique challenges in operator learning.

First, (**challenge 1**) PDE datasets demonstrate substantial variations across equation types, boundary conditions, and spatiotemporal resolutions. This diversity causes conflicting knowledge patterns when

---

*Equal contribution.
†Corresponding author.
[1]Our code is available at https://github.com/haiyangxin/MoEPOT.

39th Conference on Neural Information Processing Systems (NeurIPS 2025).

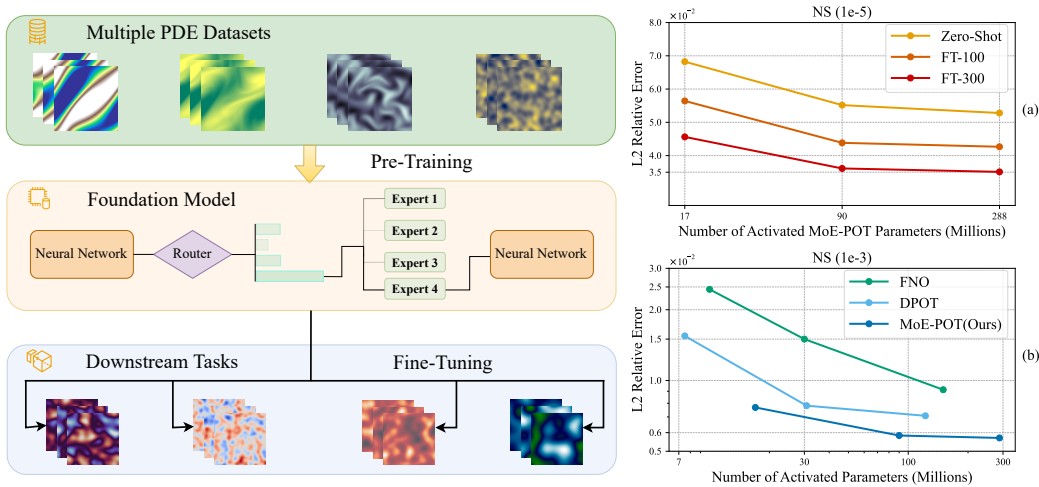

Figure 1: **Left.** An illustration of pre-training a PDE foundation model using extensive data from diverse datasets. The pre-trained model is subsequently fine-tuned for various downstream operator learning tasks, enabling the handling of complex scenarios. **Right. (a)** Comparison of errors across different numbers of fine-tuning epochs; **(b)** Comparison of zero-shot errors across different models.

merging different PDE types during training—direct data mixing frequently results in detrimental interference that restricts knowledge acquisition, rather than enhancing model generalization.

Second, (**challenge 2**) existing approaches to scaling model capacity through dense architectural expansion (increasing width/depth) incur prohibitive inference costs, making pre-training impractical for real-world applications.

To address these challenges, we propose the **M**ixture-**o**f-**E**xperts **P**re-training **O**perator **T**ransformer (**MoE-POT**), a novel sparse architecture. Our key insight is to decouple capacity expansion from computational cost through dynamic expert activation.

Specifically, MoE-POT employs a learnable router-gating network that automatically selects 4 routed experts based on each layer's input data, working with 2 fixed shared experts. The outputs from these 6 experts are then weighted and aggregated to produce the final result. The router-gating network dynamically selects routed experts that specialize in learning distinctive features of the current PDE category, effectively isolating interference from significantly different PDE data types. Meanwhile, the shared experts act as fixed computational modules for all data. They ensure consistent learning of fundamental dynamic evolution laws.

The key contributions and advantages of MoE-POT are summarized as follows:

- **Novel Architecture**: We introduce MoE-POT, a novel sparse architecture for neural operator pre-training. MoE-POT employs two types of expert networks: routed experts and shared experts, enabling a balance between generalization and specialization.

- **Empirical Validation**: As shown in Figure 1 (right), we pre-train models on 6 public PDE datasets and design multiple versions of MoE-POT with total parameter scales ranging from 30M to 0.5B. Our model with 90M activated parameters achieves up to a 40% reduction in zero-shot error compared to existing models with 120M activated parameters.

- **Interpretable**: We observe that the trained router-gating network can infer the PDE type of input data with 98% accuracy, showing MoE-POT's ability to effectively handle diverse PDE datasets. This result validates both the rationale and effectiveness of the MoE architecture.

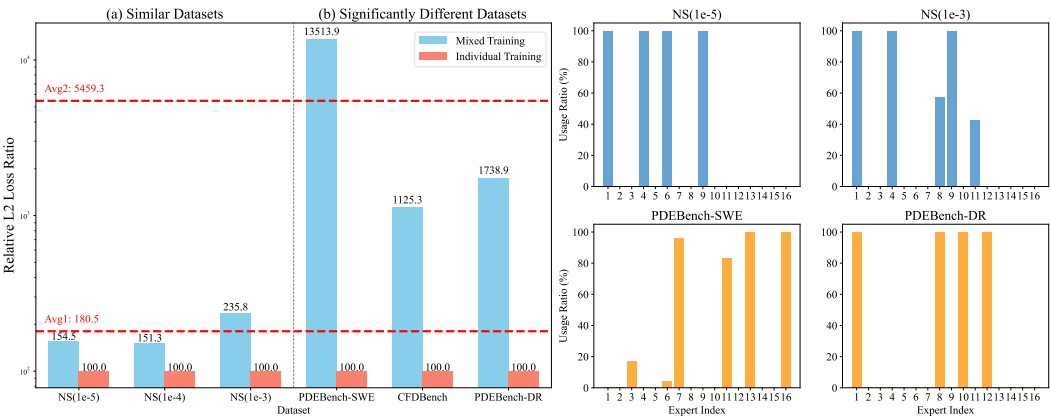

Figure 2: **Left.** The impact of mixed training on multiple datasets using FNO on model performance. **Right.** Usage ratio of routed experts in different datasets in block 4.

## 2 Preliminaries

### 2.1 Time-dependent PDE Problem

We consider a general form of parameterized time-dependent PDEs characterized by variables $\boldsymbol{u}(x,t) \in \mathbb{R}^m$. These equations satisfy the following conditions:

$$\frac{\partial \boldsymbol{u}}{\partial t} - \mathcal{F}[\boldsymbol{u};\theta](x,t) = 0, \qquad (x,t) \in \Omega \times T \subset \mathbb{R}^{d+1}, \tag{1}$$

$$\boldsymbol{u}(x,0) = \boldsymbol{u}^0(x), \quad x \in \Omega, \qquad \mathcal{B}[\boldsymbol{u}](x,t) = 0, \quad x \in \partial\Omega.$$

Here, $\mathcal{F}[\boldsymbol{u};\theta](x,t) = F(t,x,\boldsymbol{u},\partial_x\boldsymbol{u},\partial_{xx}\boldsymbol{u},\ldots;\theta)$ represents a differential operator involving spatial derivatives, while $\theta \in \boldsymbol{\Theta}$ denotes unknown parameters that define the type and coefficients of the PDE. The initial condition is given by $\boldsymbol{u}^0(x)$, and $\mathcal{B}[\boldsymbol{u}](x,t)$ specifies the boundary conditions. This general formulation encompasses a variety of fundamental PDEs.

In practical scenarios, datasets are often collected from multiple PDEs, represented as $\mathcal{D} = \cup_{k=1}^K \mathcal{D}_k$, where $\mathcal{D}_k = \{\boldsymbol{u}_i\}_{1 \le i \le N_k}$. Each solution function $\boldsymbol{u}_i \in \mathcal{D}$ is discretized on spatiotemporal meshes, expressed as $\boldsymbol{u}_i = (\boldsymbol{u}_i^1,\ldots,\boldsymbol{u}_i^T)$, with $\boldsymbol{u}_i^t = \{(x_j,u_j^t) : x_j \in \mathcal{X}_i\}$ for $1 \le t \le T$. The spatial meshes $\mathcal{X}_i$ may consist of regular grids or irregular point clouds, depending on the geometry of the domain. The parameters $\theta$ govern the type and specific characteristics of the PDE. However, in many real-world applications, such as climate modeling, only observational trajectories of data are available, while the detailed parameters $\theta$ remain inaccessible. To predict future timesteps, it is essential to infer the most likely $\theta$ implicitly from the observed sequence of $T$ frames $(\boldsymbol{u}_i^1,\ldots,\boldsymbol{u}_i^T)$.

### 2.2 Auto-regressive Denoising Pre-training

To effectively learn from temporal PDE datasets, we propose a neural operator $\mathcal{G}_w(\boldsymbol{u}^{t<T})$, parameterized by weights $w$, which auto-regressively takes $T$ frames as input and predicts the next frame based on the previous frames:

$$\boldsymbol{u}^T = \mathcal{G}_w(\boldsymbol{u}^0,\ldots,\boldsymbol{u}^{T-1}). \tag{2}$$

By predicting the next frame, the model learns an internal representation of the underlying PDE dynamics. However, directly supervising the one-step loss has been shown to be suboptimal [3]. Following DPOT [15], we inject small-scale noise into the input frames. For $\forall t \le T$, let $\boldsymbol{u}^{<t}$ denote $(\boldsymbol{u}^0,\ldots,\boldsymbol{u}^{t-1})$, and the noise $\boldsymbol{\varepsilon}$ is sampled as $\boldsymbol{\varepsilon} \sim \mathcal{N}(0,\epsilon||\boldsymbol{u}^{<t}||I)$. Noise injection improves robustness and reduces the discrepancy between training and inference.

We adopt the experimental setup of DPOT [15] and FNO [28], which focuses on the challenging scenario where models must infer system dynamics solely from solution trajectories, without access to the governing PDE parameters. In contrast to parameter-informed approaches, our core contribution is the integration of a MoE architecture into this auto-regressive paradigm.

# 3 Motivation

## 3.1 Challenges in Dense Neural Operator Pre-training

Current dense neural operator pre-training approaches face two challenges, which severely limit the model's ability to generalize across PDE tasks and hinder performance improvement.

**(C1) Performance degradation due to dataset heterogeneity**: As shown in Figure 2 (left), our preliminary experiments reveal that the heterogeneity of PDE datasets significantly impacts pre-training effectiveness. For instance, when training on different parameter configurations within the same equation family (e.g., fluid simulation data with varying Reynolds numbers), the average test error increases by only 80% compared to training on individual datasets. However, when mixing three entirely different equation types for training, the error increases by up to 5000%.

The properties of different PDE types exhibit substantial variations, yet dense models enforce parameter sharing across all inputs, making it difficult to efficiently absorb diverse PDE knowledge within a unified architecture. For example, when simultaneously learning PDEBench-SWE and PDEBench-DR, the model must encode two fundamentally different differential operators within the same parameter space, leading to negative transfer or inter-task interference [6].

**(C2) Scaling bottlenecks in model capacity and performance**: To better capture the diverse properties of PDEs, increasing model parameters is often necessary to enhance the expressive power of neural operators. However, this approach introduces significant computational cost. As shown in Figure 1 (right), experiments demonstrate that models follow a diminishing returns scaling law: improving model performance requires a substantial increase in parameters. Since all parameters are activated during inference, dense models generate high inference costs, further exacerbating computational challenges.

## 3.2 Sparse Pre-training with Mixture-of-Experts

To address these challenges, we propose a PDE pre-training model based on the MoE architecture.

**Efficient scaling with sparse activation**: As illustrated in Figure 3, the MoE architecture decomposes the fully connected computation of traditional dense layers into a collaborative mechanism involving parallel expert networks and gated routing. Each Transformer layer consists of 16 routed experts and 2 shared experts. For a given input sample, the router-gating network activates only 4 routed experts, resulting in an actual computational cost equivalent to only 33% of the total parameters. This mechanism effectively scales model capacity while controlling inference cost (**C2**).

**Physics-driven gated routing**: The dynamic selection capability of the router-gating network provides a natural solution for integrating heterogeneous PDE knowledge (**C1**). 1. Shared experts: Through cross-task learning, the 2 shared experts are constrained to capture universal physical principles (e.g., conservation laws, symmetry). 2. Routed experts: The remaining 16 experts autonomously develop distinct functional roles to learn the unique characteristics of different PDEs.

As shown in Figure 2 (right), after training, the router-gating network decisions vary significantly across different PDE datasets. For example, NS(1e-5) and NS(1e-3), which are closely related datasets, exhibit similar gating patterns. In contrast, PDEBench-SWE and PDEBench-DR, which differ substantially, show distinct gating behaviors. More importantly, our interpretability analysis (see Experiment 5.4) reveals that gating weights can serve as identifiers for PDE types. The expert selection for a given input can be used to determine its dataset type with an accuracy of 98%. This demonstrates that the MoE architecture not only improves performance but also enables implicit equation type recognition for neural operators, paving the way for building interpretable foundational models for PDEs.

# 4 Method

**Overview**. Our proposed model architecture is illustrated in Figure 3. It begins by processing raw data through a patchification layer and a temporal aggregation layer [15], which reduces spatial-temporal resolution and extracts dynamic structures inherent to PDEs. The processed features are then passed through $N$ blocks, each of which contains a Fourier layer [13] and a MoE layer, thereby achieving efficient representation and specialization for diverse PDE tasks.

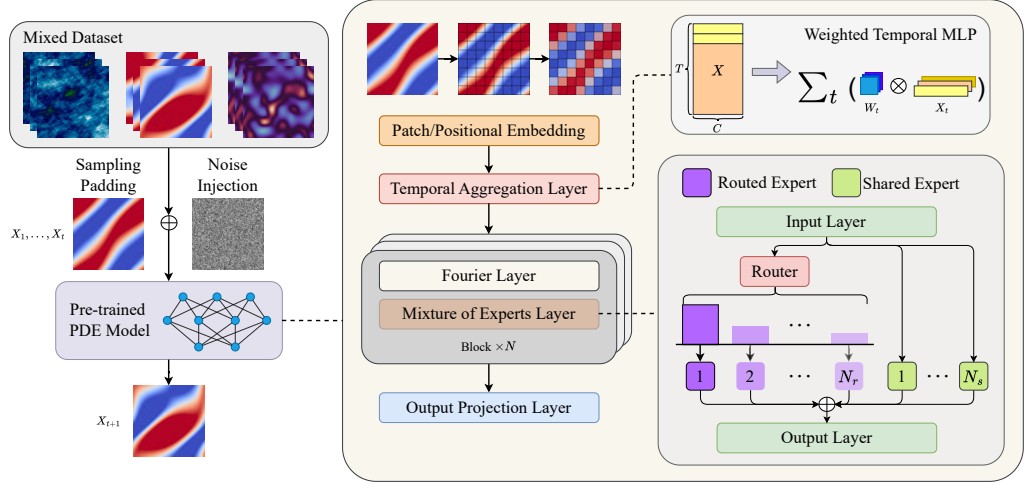

Figure 3: An illustration of our model architecture. The process begins by sampling trajectories from mixed datasets of multiple PDEs. The model is optimized by predicting the next frame based on previous frames. The mixture-of-experts layer consists of shared experts, routed experts, and a router-gating network. The router-gating network is responsible for selecting the routed experts, enabling efficient and specialized processing while ensuring scalability and modularity.

**Input Encoding and Temporal Aggregation**. The input $\boldsymbol{u}^{<T} \in \mathbb{R}^{H \times W \times T \times C}$ represents a spatiotemporal signal with $C$ channels. To encode spatial features, we apply a patchification layer with positional embeddings inspired by vision transformers [10]:

$$Z_p^t = \mathcal{P}(\boldsymbol{u}^t + \boldsymbol{p}^t), \quad t = 1, \ldots, T, \tag{3}$$

where $\mathcal{P}$ is a convolutional layer, and $p_{i,j}^t = W_p(x_i, y_j, t)$ denotes learnable positional encodings. The output $Z_p^t \in \mathbb{R}^{H/p \times W/p \times C}$ captures spatial features, with $W_p \in \mathbb{R}^{n \times 3}$, where $n$ is the feature dimension of the positional encoding (e.g. $n = C$). To capture temporal dynamics, we employ a temporal aggregation layer that extracts information across adjacent time steps. For each local node feature $\boldsymbol{z}_p^t \in \mathbb{R}^C$ in $Z_p^t$, we apply a learnable MLP transformation $W_t$ combined with Fourier feature constant $\boldsymbol{\gamma} \in \mathbb{R}^C$:

$$\boldsymbol{z}_{\text{agg}} = \sum_t W_t \cdot \boldsymbol{z}_p^t e^{-i\boldsymbol{\gamma}t}. \tag{4}$$

This aggregation enables the model to implicitly infer the underlying PDE governing parameters.

**Fourier Layer**. Using a multi-head architecture, the Fourier layer is designed to learn complex kernel-based integral transformations that approximate PDE solutions [13, 15]. Let $z^l(x)$ denote the feature at spatial location $x$ in the $l$-th block, and $Z^l$ its discretized representation. We apply a kernel integral operator $\mathcal{K}_\phi$ parameterized by a neural network:

$$(\mathcal{K}_\phi z^l)(x) = \int_\Omega \kappa(x, y; \phi) z^l(y) \mathrm{d}y, \tag{5}$$

where $\kappa(x, y; \phi)$ is a learnable kernel function. To reduce computational complexity, we constrain the kernel to be translation-invariant: $\kappa(x, y; \phi) = \kappa(x - y; \phi)$. This reformulation allows efficient implementation in the Fourier domain:

$$(\mathcal{K}_\phi z^l)(x) = \mathcal{F}^{-1}[R_\phi \cdot \mathcal{F}[z^l]]. \tag{6}$$

Here, $z^l(x) \in \mathbb{R}^{d_z}$, $R_\phi(k) \in \mathbb{C}^{d_z \times d_z}$ is a frequency-dependent learnable transformation, and $\mathcal{F}/\mathcal{F}^{-1}$ denote the Fourier transform and its inverse. To ensure memory efficiency and jointly attend to information from different representation subspaces, we first divide spatial features $z^l(x)$ into $h$ groups. The grouping is performed on the channel dimension, where $h$ is the number of heads, i.e., $z^l = \text{Concat}(z_1^l, z_2^l, \ldots z_h^l)$, where $z_i^l(k) \in \mathbb{R}^{\frac{d_z}{h}}$. we approximate $(\mathcal{K}_\phi z^l)(x)$ using $h$ smaller MLPs:

$$z_{0i}^l(x) = \mathcal{F}^{-1}[W_{2,i}^l \cdot \sigma(W_{1,i}^l \cdot \mathcal{F}[z_i^l] + b_{1,i}^l) + b_{2,i}^l](x), \tag{7}$$

where $W_{1,i}^l, W_{2,i}^l \in \mathbb{R}^{d_z/h \times d_z/h}$ and $b_{1,i}^l, b_{2,i}^l \in \mathbb{R}^{d_z/h}$ are learnable parameters, and $\sigma(\cdot)$ is an activation function. We set $z_0^l = \text{Concat}(z_{01}^l, z_{02}^l, \ldots z_{0h}^l)$, that passed to the MoE layer.

**Mixture of Experts Layer**. To facilitate sparse activation and enable expert specialization, we integrate a MoE layer specifically designed for PDE inputs. Both expert networks and router-gating networks are implemented using convolutional neural networks (CNNs) to preserve spatial information. For the reason behind the structural design, please see Appendix B.2. These features are then passed to a router-gating network $G^l(z_0^l(x))$, which computes a vector of routing logits $s^l(z_0^l(x)) \in \mathbb{R}^{N_r}$, where $N_r$ is the number of routed experts (e.g., $N_r = 16$). The gating weights are computed via a softmax function:

$$w^l(z_0^l(x)) = \text{Softmax}(s^l(z_0^l(x))) \in \mathbb{R}^{N_r}. \tag{8}$$

To maintain sparsity, only the Top-$K$ entries in $w^l(z_0^l(x))$ are retained (e.g., $K = 4$), and the rest are masked to zero:

$$\text{TopK}(w^l(z_0^l(x))) = \{(i_k, w_k^l(z_0^l(x)))\}_{k=1}^K, \tag{9}$$

where $i_k$ is the index of the $k$-th selected routed expert and $w_k^l(x)$ is the normalized routing weight.

Let $\mathcal{E}_s^l = \{E_1^{l(s)}, \ldots, E_{N_s}^{l(s)}\}$ denote the set of shared experts, which are always activated for every input. Let $\mathcal{E}_r^l = \{E_1^{l(r)}, \ldots, E_{N_r}^{l(r)}\}$ denote the set of routed experts, from which the top-$K$ are selected dynamically per input. Each expert $E_i^{l(s)}$ or $E_j^{l(r)}$ is a convolutional subnetwork that takes $z_0^l(x)$ as input and maps it to an output feature map of the same shape. Specifically, the final output of the MoE layer is computed as [8, 51]:

$$z^{l+1}(x) = \frac{1}{N_s} \sum_{i=1}^{N_s} E_i^{l(s)}(z_0^l(x)) + \sum_{k=1}^K w_k^l(z_0^l(x)) \cdot E_{i_k}^{l(r)}(z_0^l(x)). \tag{10}$$

**Load Balancing Objective.** To encourage uniform utilization of all routed experts and avoid routing collapse [51], we introduce a load balancing loss during the training phase. Following prior work [11, 8], we define the importance of each expert over a batch $\mathcal{B}$ ($|\mathcal{B}| = B$) as the sum of its routing weights:

$$\text{Importance}_i^l = \sum_{b=1}^B w_{i,b}^l(x). \tag{11}$$

We compute the coefficient of variation (CV) across all $N_r$ routed experts, and define the loss as:

$$\mathcal{L}_{\text{balance}}^l = w_{\text{bal}} \cdot \text{CV}(\{\text{Importance}_i^l\}_{i=1}^{N_r})^2, \tag{12}$$

where $w_{\text{bal}}$ is a tunable scaling factor (e.g., $w_{\text{bal}} = 0.1$). This auxiliary loss regularizes the routing distribution to maintain a balanced expert load and improves overall training stability.

**Loss Function.** The primary objective of the model is to predict the one-step transition between samples from different datasets. The loss function is defined as:

$$\mathcal{L} = \sum_{1 \leqslant t \leqslant T} \|\mathcal{G}_w(\boldsymbol{u}^{<t} + \boldsymbol{\varepsilon}) - \boldsymbol{u}^t\|_2^2 + \sum_{l=1}^N \mathcal{L}_{\text{balance}}^l, \tag{13}$$

where $\mathcal{G}_w$ represents the model's prediction function, $\boldsymbol{u}^{<t}$ denotes the input from previous timesteps, and $\boldsymbol{\varepsilon}$ is a perturbation term. By predicting the next timestep data from previous frames, the model learns to implicitly infer the PDE's governing dynamics and propagate the solution forward in time.

## 5 Experiments

We conducted comprehensive experiments to evaluate the performance of MoE-POT. This section is organized as follows: 1. Comparison with various small and pre-trained models on 6 PDE datasets. 2. Testing knowledge transfer capabilities on downstream tasks. 3. Investigating scaling laws to understand performance trends. 4. Interpretable analysis of the router-gating network selection. 5. Analyzing model inference time 6. Ablation studies to assess the impact of hyperparameters.

| Dataset | Activated Params | FNO-$\nu$ | | PDEBench | | | CFDBench |
|---|---|---|---|---|---|---|---|
| | | NS (1e-5) | NS (1e-3) | CNS(0.1,0.01) | SWE | DR | |
| Small Model | | | | | | | |
| FNO | 0.5M | **0.156** | 0.0128 | 0.170 | 0.00440 | 0.120 | 0.00761 |
| UNet | 25M | 0.198 | 0.0245 | 0.357 | 0.0521 | 0.0971 | 0.0209 |
| FFNO | 1.2M | 0.161 | 0.0256 | 0.183 | 0.00458 | 0.161 | 0.0990 |
| GK-T | 1.1M | 0.260 | 0.0148 | 0.919 | 0.0453 | **0.0120** | 0.419 |
| Oformer | 1.8M | 0.289 | **0.00319** | **0.161** | 0.00474 | 0.991 | **0.00444** |
| GNOT | 2.2M | 0.590 | 0.316 | 0.533 | **0.00199** | 0.930 | 0.0216 |
| Pre-trained | | | | | | | |
| FNO-T | 10M | 0.191 | 0.0245 | 0.0859 | 11.0 | 0.530 | 0.00601 |
| FNO-S | 30M | 0.157 | 0.0225 | 0.357 | 0.184 | 0.385 | 0.00460 |
| FNO-M | 150M | 0.141 | 0.00730 | - | 0.0104 | **0.0112** | - |
| DPOT-T | 7.5M | 0.107 | 0.0155 | 0.0168 | 0.00631 | 0.0577 | 0.00673 |
| DPOT-S | 30.8M | 0.0688 | 0.00781 | 0.0244 | 0.00392 | 0.0367 | 0.00870 |
| DPOT-M | 122M | 0.0569 | 0.00708 | 0.0224 | 0.00247 | 0.0288 | 0.0113 |
| Ours-T | 17M | 0.0682 | 0.00768 | 0.0105 | 0.00640 | 0.0411 | 0.00529 |
| Ours-S | 90M | 0.0552 | 0.00583 | 0.00959 | **0.00289** | 0.0342 | **0.00448** |
| Ours-M | 288M | **0.0528** | **0.00570** | **0.00914** | 0.00299 | 0.0300 | 0.00513 |
| Fine-tuned | | | | | | | |
| DPOT-T | 7.5M | 0.0700 | 0.00725 | 0.0168 | 0.00313 | 0.0289 | 0.00391 |
| DPOT-S | 30.8M | 0.0502 | 0.00635 | 0.0238 | 0.00315 | 0.0215 | 0.00586 |
| DPOT-M | 122M | 0.0424 | 0.00593 | 0.0221 | 0.00260 | 0.0175 | 0.00653 |
| Ours-T | 17M | 0.0456 | 0.00493 | 0.00746 | 0.00305 | 0.0188 | 0.00330 |
| Ours-S | 90M | 0.0361 | **0.00376** | **0.00742** | 0.00251 | 0.0182 | **0.00313** |
| Ours-M | 288M | **0.0351** | 0.00388 | 0.00744 | **0.00193** | **0.0140** | 0.00398 |

Table 1: Results of main experiments are divided into three parts. We use L2RE as the evaluation metric, where lower L2RE indicates better performance. We **bold** the best results in each part. We highlight the globally best results using blue . '-' indicates the error is greater than 20, signifying that failed completely. The first part is trained and evaluated individually on each dataset, the second part shows zero-shot results, and the last part shows results for fine-tuning on each dataset.

**Datasets.** For pre-training, we utilize 6 datasets sourced from 3 benchmark collections: FNO [28], PDEBench [53], and CFDBench [38]. These datasets encompass a wide range of PDE types and parameters. The mathematical formulations of these PDEs are provided in Appendix B.5. To ensure consistency and compatibility across datasets, we applied preprocessing techniques such as padding and masking. Detailed descriptions of preprocessing steps can be found in Appendix B.1.

**Training and Evaluation.** The model configurations for different scales are detailed in Appendix B.3. Across all model sizes, we employed the Adam optimizer with a learning rate of $1 \times 10^{-3}$ and trained the models for 1000 epochs. Training was conducted on servers equipped with 8 RTX 4090 GPUs, each with 24 GB of memory. We use the $l_2$ relative error (L2RE) as the primary metric to evaluate prediction quality, following the standard practice outlined in [28].

**Baseline.** We selected the following influential methods as baselines for comparison, categorized into two groups: 1. **Small models**: This group includes FNO (along with Geo-FNO for irregular datasets) [28, 26], UNet [50], FFNO [55], GK-Transformer [5], OFormer [27], and GNOT [16]. These models are trained and evaluated individually on each dataset.

2. **Pre-trained models**: FNO-(T/S/M): Larger-scale variants of the original FNO model, with expanded parameter sizes for comparison; DPOT-(T/S/M): The state-of-the-art pre-trained neural operator model [15]. Both pre-trained baselines and MoE-POT-(T/S/M) are first pre-trained on six datasets and then evaluated for performance.

## 5.1 Main Experiments

Table 1 summarizes the results of our main experiments. The parameter in the second row corresponds to the PDE dataset configuration, such as $1e-5$ for viscosity in the FNO NS dataset [28]. The activation parameter counts for MoE-POT-(T/S/M) are 17M, 90M, and 188M, respectively, with total parameter counts of 30M, 166M, and 489M.

The second part of the Table 1 evaluates pre-trained models, including FNO variants, DPOT, and MoE-POT. Our model achieves the best zero-shot performance on 5 out of 6 datasets, with significant improvements in L2RE compared to DPOT and FNO-M. For example, on the PDEBench-CNS(0.1, 0.01), Ours-S (with 90M activation parameters) reduces L2RE by 57% compared to DPOT-M (with 122M activation parameters). The FNO architecture, not specifically designed for pre-training, struggles to optimize datasets with highly diverse properties due to its dense network structure. This leads to instability during pre-training, resulting in large errors or training collapse on certain datasets. For instance, FNO-M fails to produce valid results on PDEBench-CNS(0.1, 0.01) and CFDBench due to excessively high L2RE. The performance improvements of our model are primarily attributed to the expert network design within the MoE architecture, which effectively captures intrinsic features across datasets with differing properties without mutual interference. This demonstrates the effectiveness of the MoE architecture in pre-training scenarios for PDEs.

The last part of the Table 1 presents the results of fine-tuning pre-trained models on each subset for 200 epochs. Fine-tuning consistently improves performance across all datasets, with larger models yielding better results. For example, MoE-POT-M achieves the best fine-tuning results on 5 out of 6 datasets, reducing L2RE by over 50% compared to the zero-shot model on PDEBench-DR. Compared to DPOT, our MoE-based model achieves significant performance gains. These improvements stem from the ability of the MoE architecture to substantially expand the total model parameters while keeping activation parameters relatively constant, thereby enhancing model performance without increasing inference costs. Additionally, after fine-tuning, our model outperforms all small models on most datasets, achieving state-of-the-art results on 4 out of 6 datasets. This demonstrates that our model successfully learns from multiple PDE datasets simultaneously through pre-training.

In summary, fine-tuning significantly enhances performance, suggesting that pre-training on large-scale PDE datasets is a promising and scalable approach for improving operator learning tasks. The results highlight the advantages of our MoE architecture in handling complex and heterogeneous PDE data. Furthermore, for additional comparative experiments with large-scale models, including DPOT-L and Poseidon, please refer to Appendix C.2.

## 5.2 Downstream Tasks Experiments

To evaluate the effectiveness of our pre-trained model in enhancing performance across diverse PDE downstream tasks, we conducted experiments to test its broader applicability. We selected three downstream tasks. NS ($1e$-4), closely related to the pre-training datasets NS ($1e$-3), and PDEArena, which represents equations with entirely different mathematical structures. All models were trained or fine-tuned for 500 epochs, and the results are summarized in Table 2.

Firstly, in all experiments, our model trained from scratch ('w/o Pre-train') consistently outperforms smaller models, demonstrating the effectiveness of the MoE-POT architecture for operator learning. Secondly, across all tasks, both DPOT and MoE-POT models show significant performance improvements after pre-training and fine-tuning, far surpassing the performance of small models

| Dataset | Geo-FNO | U-Net | FFNO | DPOT | | Ours | |
| | | | | w/o Pre-train | w/ Pre-train | w/o Pre-train | w/ Pre-train |
|---|---|---|---|---|---|---|---|
| NS (1e-4) | 0.107 | 0.413 | 0.220 | 0.0599 | 0.0264 | 0.0480 | **0.0160** |
| CNS (1, 0.01) | 0.0813 | 0.0827 | 0.390 | 0.0521 | 0.0398 | 0.0381 | **0.0307** |
| PDEArena | 0.154 | 0.167 | 0.161 | 0.111 | 0.0621 | 0.137 | **0.0618** |

Table 2: Experimental results of fine-tuning on downstream tasks. L2RE is used as the evaluation metric, where lower values indicate better performance. The first column shows the challenges associated with each downstream task. We **bold** the best results. "w/ Pre-train" refers to fine-tuning after pre-training, while "w/o Pre-train" refers to training from scratch.

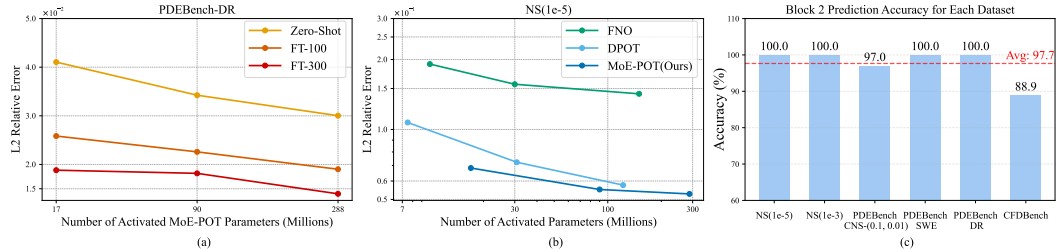

Figure 4: **(a)** Comparison of errors across different numbers of fine-tuning epochs; **(b)** Comparison of zero-shot errors across different models; **(c)** Dataset classification accuracy based on router-gating network selection.

trained from scratch. This indicates that pre-training enables the model to learn more effective and transferable representations. These results highlight the remarkable versatility of our model, which can be seamlessly extended to a wide range of downstream tasks.

### 5.3 Scaling Experiments

The relationship between performance and increasing model size is a critical property of pre-trained models. In this study, we conduct scaling experiments to evaluate the scalability of our model. The results are presented in Figure 4 (a). We observe that as the model size increases, the zero-shot test error consistently decreases, approximately following a scaling law. Furthermore, fine-tuning the model on specific datasets leads to improved performance. As shown in Figure 4 (b), while all models exhibit scaling properties, our model demonstrates better performance for a given number of activated parameters. This advantage primarily stems from the MoE architecture, which significantly increases the total number of parameters without proportionally expanding the number of activated parameters. For example, MoE-POT-T with a total of 30M parameters, requires only 57% of the activated parameters (17M) compared to existing models with similar performance. Furthermore, further analytical experiments are presented in Appendices C.3, C.4, and C.5, which include an analysis of error accumulation over rollout steps, a study on the relationship between fine-tuning data size and performance, and an investigation into the impact of pre-training data heterogeneity.

### 5.4 Interpretable Analysis

We aim to determine which dataset a given data point belongs to by analyzing the expert selection in the MoE router-gating network. For a specific block, we first compute the average expert selection values $Y_i$ for each dataset (forming a vector), where $i = 1, \ldots, 6$ represents the $i$-th dataset. Then, for the expert selection vector $I_0$ of input, we calculate its distance to each $Y_i$. The $Y_{i_0}$ with the smallest distance to it is obtained, indicating that the input belongs to the $i_0$-th dataset. Detailed procedures can be found in Appendix B.4.

As shown in Figure 4 (c), the router-gating network in Block 2 achieves an accuracy of 97.7% in classifying the input dataset. Similar results are observed in other blocks. This strongly demonstrates that the MoE architecture effectively learns the differences between PDE datasets and uses this information for classification. Furthermore, further interpretability analysis, including the emergence of classification ability and its generalization to out-of-distribution (OOD) data, is available in Appendix C.6.

| Model | DPOT | | | | Ours | | |
|---|---|---|---|---|---|---|---|
| | Tiny | Small | Medium | Large | Tiny | Small | Medium |
| Activated Parameters (M) | 7.5 | 30 | 158 | 493 | 17 | 90 | 288 |
| Total Parameters (M) | 7.5 | 30 | 158 | 493 | 30 | 166 | 489 |
| Inference Time (ms) | 5.5 | 6.5 | 16.7 | 24.3 | 8.8 | 12.7 | 16.6 |

Table 3: Average single-step inference time of different models on the NS $(1e - 5)$ dataset.

| $N_r$ | NS(1e-3) | NS(1e-5) | CNS(0.1,0.01) | SWE | DR | CFDBench | Top-$K$ | NS(1e-3) | NS(1e-5) | CNS(0.1,0.01) | SWE | DR | CFDBench |
|---|---|---|---|---|---|---|---|---|---|---|---|---|---|
| 32 | **0.06680** | 0.00857 | **0.01011** | 0.00440 | 0.04384 | 0.00630 | 4 | **0.06920** | **0.00762** | **0.01046** | 0.00639 | **0.04094** | 0.00663 |
| 16 | 0.06920 | **0.00762** | 0.01046 | 0.00639 | **0.04094** | 0.00663 | 2 | 0.06983 | 0.00777 | 0.01157 | **0.00439** | 0.04142 | **0.00570** |
| 8 | 0.06833 | 0.00773 | 0.01035 | **0.00417** | 0.04153 | **0.00571** | 1 | 0.10108 | 0.01379 | 0.02734 | 0.00781 | 0.07896 | 0.00968 |

Table 4: Results of ablation experiments on the influences of the number of routed experts $N_r$ (left part) and the number of expert selections Top-$K$ (right part). L2RE is used as the evaluation metric.

## 5.5 Inference Time Analysis

We evaluated the inference time of various models. As shown in Table 3, under the same total number of parameters, our model demonstrates significantly lower inference time compared to DPOT. For example, MoE-POT-M has a total parameter count of 489M, comparable to DPOT-L, yet its inference time is only 68% of the latter, making it equivalent to DPOT-M with just 158M parameters. This efficiency is primarily attributed to the MoE structure, which activates far fewer parameters than the total parameter count. For certain PDE tasks, a single computation may require $10^3$ to $10^5$ inference steps. The MoE structure effectively reduces inference time while preserving model performance.

## 5.6 Ablation Experiments

We conduct ablation studies by training MoE-POT-T on 6 pre-trained datasets and compare the averaged zero-shot performance on the corresponding test datasets. As shown in Table 4, the error remains stable when the number of routed experts $N_r$ is sufficiently large. However, increasing $N_r$ leads to higher computational costs during training. Thus, we select $N_r = 16$ as a balance between performance and efficiency. Increasing the number of expert selections (Top-$K$) reduces error, as more experts contribute to inference, thereby increasing the activated parameters. However, this improvement exhibits diminishing returns, so we set Top-$K = 4$ as an optimal trade-off. Additionally, we analyzed the impact of the number of heads $h$ and patch sizes on the model's performance. Please refer to Appendix C.1 for details.

# 6 Limitations and Conclusions

This paper introduces MoE-POT, a sparse architecture designed for PDE pre-training. By dynamically activating routed experts and leveraging fixed shared experts, MoE-POT achieves state-of-the-art performance under the same activated parameters. Additionally, we observe that the router-gating network can effectively distinguish features of different PDEs and classify data, further illustrating the rationality of the MoE structure.

However, we have yet to analyze the mathematical essence of this classification mechanism. Exploring how PDE classification can guide the construction of more effective pre-training datasets remains an important direction for future work.

# Acknowledgements

The authors would like to thank all the anonymous reviewers for their insightful comments and valuable suggestions. This work was supported by Smart-Grid National Science and Technology Major Project under contract 2025ZD0805500, the National Key R&D Program of China under contract 2022ZD0119801, and the National Nature Science Foundations of China grants 124B1019, U23A20388, 62021001.

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

# A  Related Work

## A.1  Neural Operators

Neural operators have emerged as powerful tools for learning solution operators of partial differential equations (PDEs) directly from data, demonstrating immense potential across diverse fields such as fluid dynamics [28], thermodynamics [63], electromagnetism [1], and climate forecasting [45]. The wide-ranging mathematical properties and data formats associated with PDEs have driven substantial research into designing effective neural operator architectures.

For instance, DeepONet [36] employs a dual-network structure consisting of a branch and trunk network, while the Fourier neural operator (FNO) [28] introduces a frequency-domain approach to efficiently learn solution mappings. Building on FNO, extensions such as Geo-FNO [26], NUNO [35], and GINO [29] adapt the method to handle complex geometries. Other works [3, 41] focus on time-dependent PDEs, addressing next-time prediction challenges with specialized training strategies. Hybrid approaches like PINO [30] and PI-DeepONet [58] integrate operator learning with physics-informed neural networks (PINNs) [49, 23, 57, 7], leveraging physical constraints to enhance generalization and reduce reliance on large datasets.

Further advancements have been achieved through transformer-based architectures [5, 27, 16, 59, 44], which incorporate techniques like patchification and linear attention mechanisms. For example, the GK-Transformer [5], OFormer [27], and GNOT [16] demonstrate strong performance on problems involving irregular geometries. AFNO [13] combines the efficiency of Fourier transforms with attention mechanisms, inspired by FNO, to achieve low memory and computational costs akin to MLP-Mixer [54]. This approach has been further adapted for large-scale applications such as climate forecasting [45, 61].

Despite these advancements, existing neural operator methods often require task-specific training and large amounts of domain-specific data, underscoring the need for more data-efficient approaches to broaden their applicability and scalability.

## A.2  Pre-training in Scientific Machine Learning

Pre-training has emerged as a highly effective paradigm for enhancing downstream tasks by training models in a (self-)supervised manner on large-scale datasets. This approach has achieved remarkable success in traditional domains such as natural language processing [47, 48, 4] and computer vision [19, 18], and is increasingly showing promise in scientific machine learning applications, including protein modeling [21], molecular representation learning [64], and climate and weather modeling [43, 42, 45, 31, 32, 33, 46, 24, 34, 25, 17, 39].

In the context of learning PDE data, initial efforts have been made to explore pre-training across various physical systems [56, 9, 37]. For instance, [52] designs a relatively universal PDE model to collectively train data from multiple steady-state PDEs. [60] utilizes the MathGPT architecture to investigate in-context learning capabilities for PDE data. Additionally, MPP [40] introduces an auto-regressive approach for pre-training on time-dependent PDE datasets. [15] proposes an auto-regressive denoising pre-training strategy combined with a scalable Fourier-based model architecture, enabling efficient large-scale pre-training on PDE data.

Additionally, works such as Poseidon [20], which are based on a multiscale operator transformer, have achieved excellent pre-training effectiveness and generalization by employing a novel training strategy that leverages the semi-group property of time-dependent PDEs. These approaches primarily focus on mapping parameters to PDE solutions, which contrasts with the auto-regressive solution method discussed in this paper, giving each methodology a distinct scope of application.

However, these works primarily rely on dense neural network architectures and do not explicitly consider the relationships between different PDE datasets, which could significantly impact model performance. There remains substantial room for exploration in pre-training models for more complex scenarios and larger parameter spaces.

# B    Details of Experiment Settings

## B.1    Data Preprocessing and Sampling

We adopt the data preprocessing strategy proposed in DPOT [15], with modifications to ensure compatibility across diverse PDE datasets.

**Data Padding and Masking.** To standardize spatial resolution, we fix the resolution at $H = 128$, which aligns with a significant portion of the datasets. For datasets with lower resolutions, we upscale them to $H$ using interpolation. For datasets with higher resolutions, we downscale them to $H$ using random sampling or interpolation.

To unify the number of variables (i.e., channels) across different PDEs, we pad all datasets along the channel dimension to match the dataset with the maximum number of channels, filling unused entries with a constant value (e.g., 1). For datasets with irregular geometric shapes, we use an additional mask channel that encodes the geometric configuration of each PDE instance. This ensures consistent representation across datasets while preserving unique structural information.

**Noise inserting.** During the training process, we add noise to improve the stability of the training. We only insert noise during the pre training process, and do not insert noise in fine-tuning or downstream tasks. The insertion method of noise is as follows. For $\forall t \leq T$, denote $\boldsymbol{u}^{<t}$ as $(\boldsymbol{u}^0, \ldots, \boldsymbol{u}^{t-1})$ and the noise as $\boldsymbol{\varepsilon} \sim \mathcal{N}(0, \epsilon || \boldsymbol{u}^{<t} || I)$. Then the input is $\boldsymbol{u}^{<t} + \boldsymbol{\varepsilon}$.

**Balanced Data Sampling.** To balance the contribution of datasets with varying sizes, we assign an importance weight $w_k$ to each dataset. Let $|\mathcal{D}_k|$ denote the number of data points in the $k$-th dataset, where $1 \leqslant k \leqslant K$. The probability of sampling a data point from the $k$-th dataset is computed as:

$$p_k = \frac{w_k}{K|\mathcal{D}_k| \cdot \sum_k w_k}.$$

This sampling strategy ensures that datasets with fewer samples or higher importance scores are appropriately represented during training.

**Patchification Layer.** We follow the patch-based tokenization strategy used in Vision Transformers [10]. Given a spatiotemporal input tensor $\mathbf{u}^{<T} \in \mathbb{R}^{H \times W \times T \times C}$, we apply a convolutional embedding layer with kernel size $P \times P$ and stride $P$. This partitions the spatial domain into non-overlapping patches of size $P \times P$. Each patch is mapped to a $d$-dimensional embedding vector via a shared linear projection implemented as a convolution:

$$\text{Conv2D}(C \rightarrow d, \text{kernel} = P, \text{stride} = P).$$

This process generates a sequence of patch tokens for each timestep, which are subsequently fed into the downstream attention layers. By reducing spatial resolution while preserving local structure, this patchification approach enables efficient global modeling in the transformer blocks.

## B.2    Design of the MoE Structure

As detailed in the Section 4, our MoE architecture is built upon a CNN framework and utilizes both shared and routed experts in parallel. The design of the current MoE-POT architecture is the culmination of extensive experimentation. We explored numerous configurations, including combinations of MoE with other neural operators (such as MPP, FNO, and FFNO) and alternative MoE structural designs (e.g., omitting shared experts or employing MLP-based experts instead of CNNs). Many of these alternative designs proved unstable, yielding models with substantial errors on certain datasets that could not be corrected through fine-tuning. The final MoE-POT architecture presented in this work is the robust and effective design that emerged from this rigorous development process.

## B.3    Model sizes and training details

**Pre-training.** We selected 3 models of varying sizes, i.e., MoE-POT-Tiny, MoE-POT-Small, and MoE-POT-Medium. The specific parameters of these models are shown in Table 5. For the pre-training stage, we set the learning rate to $1 \times 10^{-3}$ and used a One-cycle learning rate schedule over 1000 epochs, with the first 200 epochs as the warm-up phase. The Adam optimizer was employed with a weight decay of $1 \times 10^{-6}$ and momentum parameters $(\beta_1, \beta_2) = (0.9, 0.9)$. Training was

conducted using 8 RTX 4090 GPUs, with a total batch size of 20. The patch size was set to 8. We set the weight $w = 1$ for all datasets. For our training process, we selected $T = 10$ timesteps to predict the next frame, aligning with the original settings of most datasets.

| Size | Attention dim | MLP dim | Layers | Heads | Routed experts | Shared experts | Top-K | Model size | Activated size |
|------|------|------|------|------|------|------|------|------|------|
| Tiny | 512 | 512 | 4 | 4 | 16 | 2 | 4 | 30M | 17M |
| Small | 1024 | 1024 | 6 | 8 | 16 | 2 | 4 | 166M | 90M |
| Medium | 1024 | 2048 | 8 | 8 | 16 | 2 | 4 | 489M | 288M |

Table 5: Configurations of MoE-POT with different sizes.

**Fine-tuning.** Our model supports fine-tuning across various downstream datasets while retaining the generalization capability learned during pretraining. Specifically, we freeze the parameters of the router-gating network during fine-tuning to preserve the expert assignment strategy obtained from the joint training stage. This strategy allows the model to reuse the learned routing behavior, enabling different experts to specialize in different data distributions. Only the expert networks are updated to adapt to the target dataset, while the router-gating network continues to provide consistent and stable expert selection. This separation of routing and expert adaptation ensures more stable and efficient fine-tuning, particularly when transferring to tasks with limited data. For the fine-tuning stage, we set the learning rate to $1 \times 10^{-3}$ and used a one-cycle learning rate schedule over 200 epochs, with the first 40 epochs as the warm-up phase. And for the downstream tasks, we set the learning rate to $1 \times 10^{-3}$ and used a one-cycle learning rate schedule over 500 epochs, with the first 100 epochs as the warm-up phase.

**Dataset size.** The train and test dataset sizes used in the pre-training and fine-tuning stages are shown in Table 6. And the train and test dataset sizes for downstream tasks are shown in Table 7. It should

| Size | FNO(1e−5) | FNO(1e−3) | CNS(0.1, 0.01) | SWE | DR | CFDBench |
|------|------|------|------|------|------|------|
| Train | 1000 | 1000 | 9000 | 900 | 900 | 9000 |
| Test | 200 | 200 | 200 | 60 | 60 | 1000 |
| Fine-tuning | 1000 | 1000 | 9000 | 900 | 900 | 9000 |

Table 6: Dataset size in pre-training and fine-tuning

be noted that NS (1e-4) and CNS (1,0.01) have similar datasets in the pre-training dataset, while pdearea differs significantly from the pre-training dataset.

| Size | NS(1e−4) | CNS(1, 0.01) | PDEArena |
|------|------|------|------|
| train | 2000 | 2000 | 2000 |
| test | 200 | 200 | 200 |

Table 7: Dataset size in downstream

**Details of inference.** To learn from temporal PDE datasets, our network $\mathcal{G}_w(\boldsymbol{u}^{t<T})$ parameterized by weights $w$ that auto-regressively takes $T$ frames as input and decodes the next frame from previous frames,

$$\boldsymbol{u}^{i+T} = \mathcal{G}_w(\boldsymbol{u}^i, \ldots, \boldsymbol{u}^{i+T-1}) \quad \forall i.$$

By predicting the next frame, we can infer the numerical solution of the final time step based on auto-regression. For example, if we take the first 10 steps as our input, we can predict the solution $x_{pred}$ for the next 10 steps. And the ground truth is $x_{gt}$, then the loss is

$$\text{Rel-}\ell_2 = \frac{\|x_{\text{pred}} - x_{\text{gt}}\|_2}{\|x_{\text{gt}}\|_2}.$$

## B.4 Interpretable Analysis Algorithms

To further investigate the router-gating network within the MoE structure, we designed the following experiment. Our goal is to leverage the section of the router-gating network to determine which dataset the input data belongs to.

Specifically, given an input sample $X$, the gating network outputs a probability vector $Y \in \mathbb{R}^{16}$ representing the likelihood of selecting each of the 16 experts. Although only the top-4 experts are used during inference, the full softmax output encodes meaningful distributional information about expert preferences.

For a specific block, we compute the average expert selection distribution $Y_i = \frac{1}{N_i} \sum_{j=1}^{N_i} Y_{ij}$, and $Y_i, Y_{ij} \in \mathbb{R}^{16}$, where $N_i$ is the number of samples from $i$-th dataset, $i = 1, ..., 6$. $Y_{ij}$ is the router-gating network output for the $j$-th sample in $i$-th dataset. Then, for any new input $X$, we compare its expert distribution $I_0 = (I_{0,1}, ..., I_{0,16})$ to all $Y_i = (Y_{i,1}, ...., Y_{i,16})$ using the cross-entropy loss function:

$$f(I_0, Y_i) = -\sum_{k=1}^{16} I_{0,k} \log(Y_{i,k}),$$

Suppose $i_0$ represents the nearest dataset.

$$i_0 = \arg\min_i f(I_0, Y_i).$$

In this case, we classify the input data $X$ as belonging to the $i_0$-th dataset.

## B.5 Mathematical Forms of Datasets

Here, we list the PDEs of the datasets we used for pre-training.

- **FNO-$\nu$ [28]:** The quantity of interest (QoI) is the vorticity $w(x,t), (x,t) \in [0,1]^2 \times [0,T]$ and it satisfies, $\nu$ represents the viscosity coefficient. In paper are FNO (1e-3), FNO (1e-4) and FNO (1e-5).

$$\partial_t w + u \cdot \nabla w = \nu \Delta w + f(x),$$
$$\nabla \cdot u = 0.$$

- **PDEBench-CNS($\eta, \zeta$) [53]:** We need to predict the velocity, pressure, and density fields $\boldsymbol{u}(x,t), p(x,t), \rho(x,t)$ where $(x,t) \in [0,1]^2 \times [0,1]$. The PDEs are as follows .$\eta$ is dynamic shear viscosity and $\zeta$ is bulk viscosity.In paper are CNS (0.1,0.01), CNS (1,0.01).

$$\partial_t \rho + \nabla \cdot (\rho \boldsymbol{u}) = 0,$$
$$\rho(\partial_t \boldsymbol{u} + \boldsymbol{u} \cdot \nabla \boldsymbol{u}) = -\nabla p + \eta \Delta \boldsymbol{u} + (\varsigma + \eta/3)\nabla(\nabla \cdot \boldsymbol{u}),$$
$$\partial_t \left(\frac{3}{2}p + \frac{\rho u^2}{2}\right) = -\nabla \cdot \left(\left(\varepsilon + p + \frac{\rho u^2}{2}\right)\boldsymbol{u} - \boldsymbol{u} \cdot \sigma'\right).$$

- **PDEBench-SWE [53]:** We need to predict water depth $h(x,t)$ where the domain is $[-1,1]^2 \times [0,5]$. The PDEs are as follows,In paper is SWE.

$$\partial_t h + \nabla \cdot (h\boldsymbol{u}) = 0,$$
$$\partial_t(h\boldsymbol{u}) + \nabla \cdot \left(\frac{1}{2}h\boldsymbol{u}^2 + \frac{1}{2}g_r h^2\right) = -g_r h \nabla b.$$

- **PDEBench-DR [53]:** We need to predict the density fields $\boldsymbol{u}(x,t)$. The domain is $[-2.5, 2.5]^2 \times [0,1]$ and the PDEs are as follows,in paper is DR

$$\partial_t \boldsymbol{u} = \boldsymbol{D}\nabla^2 \boldsymbol{u} + \boldsymbol{R}(\boldsymbol{u}).$$

- **PDEArena-NS1/2 [14]:** We need to predict the velocity, pressure, and density fields $\boldsymbol{u}(x,t), p(x,t), \rho(x,t)$ where $(x,t) \in [0,32]^2 \times [0,24]$. The PDEs are as follows,in paper is PDEArena.

$$\partial_t \boldsymbol{v} = -\boldsymbol{v} \cdot \nabla \boldsymbol{v} + \mu \nabla^2 \boldsymbol{v} - \nabla p + \boldsymbol{f},$$
$$\nabla \cdot \boldsymbol{v} = 0.$$

- **CFDBench [38]:** We need to predict the velocity and pressure fields $\boldsymbol{u}(x,t), p(x,t)$. The domains are different as this is a dataset with irregular geometries. The PDEs are as follows, in paper is CFDBench.

$$\partial_t(\rho\boldsymbol{u}) + \nabla \cdot (\rho\boldsymbol{u}^2) = -\nabla p + \nabla \cdot \mu(\nabla\boldsymbol{u} + \nabla\boldsymbol{u}^T),$$
$$\nabla \cdot (\rho\boldsymbol{u}) = 0.$$

# C    Experimental Data and Supplementary Experiments

## C.1    Partial Hyperparameter Ablation Experiment

| $h$ | NS(1e-3) | NS(1e-5) | CNS | SWE | DR | CFDBench | $P$ | NS(1e-3) | NS(1e-5) | CNS | SWE | DR | CFDBench |
|---|---|---|---|---|---|---|---|---|---|---|---|---|---|
| 2 | **0.06748** | 0.00760 | **0.01034** | 0.00495 | 0.04277 | 0.00559 | 4 | **0.06226** | 0.00819 | 0.01765 | **0.00396** | **0.03481** | **0.00579** |
| 4 | 0.06920 | 0.00762 | 0.01046 | 0.00639 | **0.04094** | 0.00663 | 8 | 0.06920 | **0.00762** | **0.01046** | 0.00639 | 0.04094 | 0.00663 |
| 8 | 0.06963 | **0.00709** | 0.01036 | **0.00323** | 0.04199 | **0.00538** | 16 | 0.08964 | 0.00792 | 0.01301 | 0.00673 | 0.11671 | 0.00847 |

Table 8: Results of ablation experiments on the influences of the number of heads $h$ (left part) and patch sizes $P$ (right part). L2RE is used as the evaluation metric.

Table 8 demonstrates that the number of heads $h$ has minimal impact on error but affects computational cost. Accordingly, we choose $h = 4$ for efficiency. Finally, medium patch sizes ($P = 4$ or $8$) help reduce error, leading us to select $P = 8$ for optimal performance.

## C.2    More Comparative Experiments

We selected DPOT [15] as our primary multi-physics baseline for the following reasons:

1. Clear Attribution of Gains: Our MoE-POT architecture is a direct modification of the DPOT model, where we replace the dense feed-forward network with our proposed sparse MoE layer. This controlled comparison allows us to cleanly attribute any performance improvements directly to the MoE architecture, providing a clear and rigorous validation of our core contribution.

2. Divergent Experimental Paradigms: Our work, following DPOT, operates under an auto-regressive paradigm, predicting future states based solely on a sequence of previous solution frames. In contrast, models like Poseidon [20] and MPP [40] are designed for a parameter-informed setting, where they take explicit problem parameters (e.g., coefficients, boundary conditions) as input to predict a future state. The public benchmark datasets used in our primary experiments (from FNO, PDEBench, and CFDBench) do not provide these explicit PDE parameters, making a direct comparison with parameter-informed models infeasible under our main experimental protocol.

This section provides additional experimental results and analysis to supplement the main paper. We compare our MoE-POT architecture with the larger DPOT-L model and the Poseidon.

### C.2.1    Experimental Results with DPOT-L

To provide a more comprehensive comparison against large-scale dense models, we evaluated the performance of DPOT-L (493M parameters). The results, alongside our MoE-POT models and smaller DPOT variants, are presented in Table 9.

| Model & Activated Params | NS(1e-5) | NS(1e-3) | CNS(0.1,0.01) | SWE | DR | CFDBench |
|---|---|---|---|---|---|---|
| DPOT-S (31M) | 0.0688 | 0.0078 | 0.0244 | 0.0039 | 0.0367 | 0.0087 |
| DPOT-M (122M) | 0.0569 | 0.0071 | 0.0224 | 0.0025 | 0.0288 | 0.0113 |
| DPOT-L (493M) | 0.0576 | 0.0061 | 0.0113 | 0.0023 | 0.0219 | 0.0065 |
| MoE-POT-T (17M) | 0.0682 | 0.0077 | 0.0105 | 0.0064 | 0.0411 | 0.0053 |
| MoE-POT-S (90M) | 0.0552 | 0.0058 | 0.0096 | 0.0029 | 0.0342 | 0.0045 |
| MoE-POT-M (288M) | 0.0528 | 0.0057 | 0.0091 | 0.0030 | 0.0300 | 0.0051 |

Table 9: Zero-shot L2 Relative Error (L2RE) comparison, including DPOT-L.

The results in Table 9 lead to two key observations: 1. Diminishing returns for dense models: The performance of the dense DPOT architecture shows diminishing returns with scale. The improvement from DPOT-M (122M) to DPOT-L (493M)—a $4\times$ increase in parameters—is marginal on several datasets (e.g., NS(1e-3)) and modest on others. 2. Competitive performance with higher efficiency: When comparing MoE-POT-M (288M activated) with DPOT-L (493M activated), our model achieves competitive, and in some cases superior, performance. MoE-POT-M outperforms DPOT-L on three of the six datasets (NS(1e-3), CNS, CFDBench), while DPOT-L holds a slight advantage on the other three.

### C.2.2 Experimental Results with Poseidon

To address the interest in comparing with the latest models, we conducted supplementary fine-tuning experiments on two challenging downstream tasks from the Poseidon paper [20]: Wave-Layer and Wave-Gauss. We evaluated both MoE-POT and Poseidon under two distinct settings to fairly assess their capabilities.

**Setting 1: Auto-regressive (Our Native Setting)** In this setting, models predict future states using only previous solution trajectories, without access to explicit PDE parameters. The results are shown in Table 10.

| Model (Activated Params) | Wave-Layer | Wave-Gauss |
| --- | --- | --- |
| Poseidon-T (21M) | 0.29 | 0.29 |
| Poseidon-B (158M) | 0.21 | 0.24 |
| MoE-POT-T (17M) | **0.07** | **0.07** |
| MoE-POT-S (90M) | **0.05** | **0.06** |

Table 10: L2 Relative Error on downstream tasks in the auto-regressive setting. Lower is better.

**Setting 2: Parameter-Informed (Poseidon's Native Setting)** In this setting, models are provided with explicit PDE parameters as additional input. We adapted our MoE-POT model to accept these parameters to ensure a fair comparison. The results are shown in Table 11.

| Model (Activated Params) | Wave-Layer | Wave-Gauss |
| --- | --- | --- |
| Poseidon-T (21M) | **0.08** | **0.06** |
| Poseidon-B (158M) | **0.06** | **0.09** |
| MoE-POT-T (17M) | 0.11 | 0.14 |
| MoE-POT-S (90M) | 0.06 | 0.10 |

Table 11: L2 Relative Error on downstream tasks in the parameter-informed setting. Lower is better.

The results from these supplementary experiments indicate that each model excels in its native operational setting. In the auto-regressive setting (Table 10), where PDE parameters are unknown, MoE-POT significantly outperforms Poseidon. This highlights our model's strength in implicitly learning system dynamics from solution trajectories alone.

Conversely, in the parameter-informed setting (Table 11), Poseidon generally demonstrates superior performance, showcasing its effectiveness when explicit physical knowledge is available. These findings suggest that MoE-POT and Poseidon have different primary application scopes rather than one being definitively superior across all scenarios.

### C.3 Rollout Error at Different Timesteps

A key challenge in auto-regressive prediction is the accumulation of errors. Even a small improvement in single-step prediction accuracy can lead to a substantial reduction in the cumulative error over a long rollout, as prediction inaccuracies propagate through the sequence.

To illustrate this effect, we analyze the rollout error at different timesteps for the Shallow Water Equations (SWE) dataset. Table 12 compares the L2RE of DPOT-S and our MoE-POT-S at frames 50, 70, and 100 of the rollout.

| Model | Frame 50 L2RE | Frame 70 L2RE | Frame 100 L2RE | Average L2RE |
|-------|---------------|---------------|----------------|--------------|
| DPOT-S | 0.0031 | 0.0034 | 0.0051 | 0.0039 |
| MoE-POT-S | 0.0024 | 0.0026 | 0.0035 | 0.0029 |

Table 12: Illustration of error accumulation on the SWE dataset. The table shows the L2RE at specific frames during a 100-step rollout.

As demonstrated in Table 12, the error for both models increases over the rollout period, confirming the effect of error accumulation. More importantly, the performance advantage of MoE-POT-S grows significantly over time. The relative error reduction compared to DPOT-S is approximately 23% at frame 50, but this gap widens to over 31% by frame 100. This super-linear divergence underscores the critical impact of achieving lower single-step prediction error, as its benefits are amplified during long-term, multi-step rollouts.

## C.4 Analysis of Fine-Tuning Sample Efficiency

This section investigates the relationship between the number of fine-tuning samples and model performance, thereby analyzing the data efficiency of the MoE-POT architecture. We conducted few-shot fine-tuning experiments on both an in-distribution and an out-of-distribution task to demonstrate how performance scales with data availability for both MoE-POT and its dense counterpart, DPOT.

We compare the performance of MoE-POT-S and DPOT-S on two fine-tuning tasks: 1. In-Distribution Task: The NS (1e-4) dataset, which is closely related to the data used during pre-training. 2. Out-of-Distribution Task: The Wave-Layer dataset from the Poseidon [20], which represents a novel physical system.

For each task, we fine-tuned both models for 500 epochs while varying the number of available training samples, and we report the final L2 Relative Error. The results for the in-distribution and out-of-distribution tasks are presented in Table 13 and Table 14, respectively.

| Number of Samples | 16 | 32 | 64 | 128 | 512 | 2000 |
|-------------------|------|------|------|------|-------|-------|
| DPOT-S | 0.25 | 0.20 | 0.13 | 0.09 | 0.044 | 0.026 |
| MoE-POT-S | **0.20** | **0.15** | **0.11** | **0.07** | **0.040** | **0.016** |

Table 13: L2 Relative Error on the in-distribution NS (1e-4) task versus the number of fine-tuning samples. Lower values are better.

| Number of Samples | 16 | 32 | 64 | 128 |
|-------------------|------|------|------|------|
| DPOT-S | 0.41 | 0.33 | 0.26 | 0.19 |
| MoE-POT-S | **0.34** | **0.26** | **0.20** | **0.14** |

Table 14: L2 Relative Error on the out-of-distribution Wave-Layer task versus the number of fine-tuning samples. Lower values are better.

The results demonstrate a clear trend: while both models improve with more data, MoE-POT-S consistently outperforms DPOT-S across all sample sizes on both tasks. This highlights the superior data efficiency of the MoE-POT architecture. The larger capacity and specialized experts of the pre-trained MoE-POT model enable it to generalize more effectively from limited data. This means it can either achieve a target performance level with significantly fewer fine-tuning examples or deliver superior accuracy given the same amount of data.

## C.5 Performance with Increasing Dataset

A core motivation for our work is the challenge of negative transfer [6] in dense neural operators when pre-trained on a mixture of heterogeneous PDE datasets. A single, dense network struggles to learn conflicting physical laws, which can degrade performance as more diverse data is added. This section presents an experiment designed to test this hypothesis and demonstrate the robustness of the MoE-POT architecture in mitigating this issue.

To investigate the impact of increasing data heterogeneity, we pre-trained both the dense DPOT-S model and our sparse MoE-POT-S model on progressively larger and more diverse mixtures of datasets. For each experiment, the models were trained from scratch on the specified data mixture. We evaluated their zero-shot performance on the original six pre-training datasets to measure how well they retained knowledge.

The dataset mixtures were constructed as follows:

- 6 Datasets: The standard pre-training set used in our main experiments: NS(1e-5), NS(1e-3), CNS(0.1, 0.01), SWE, DR, and CFDBench.
- 10 Datasets: The base set plus four additional datasets from the DPOT paper [15]: NS(1e-4), CNS(1, 0.1), and two Navier-Stokes tasks from PDEArena.
- 12 Datasets: The 10-dataset mix plus two additional CNS variants: CNS(1, 0.01) and CNS(0.1, 0.1).

| Model (Pre-trained on) | NS(1e-5) | NS(1e-3) | CNS(0.1,0.01) | SWE | DR | CFDBench |
|---|---|---|---|---|---|---|
| *Dense Model* | | | | | | |
| DPOT-S (6 Datasets) | 0.0688 | 0.0078 | 0.0244 | 0.0039 | 0.0367 | 0.0087 |
| DPOT-S (10 Datasets) | 0.0663 | 0.0069 | 0.0224 | 0.0037 | 0.0575 | 0.0146 |
| DPOT-S (12 Datasets) | 0.0739 | 0.0079 | 0.0129 | 0.0105 | 0.0724 | 0.0075 |
| *Sparse Model (Ours)* | | | | | | |
| MoE-POT-S (6 Datasets) | 0.0552 | 0.0058 | 0.0096 | 0.0029 | 0.0342 | 0.0045 |
| MoE-POT-S (10 Datasets) | 0.0521 | 0.0053 | 0.0085 | 0.0029 | 0.0371 | 0.0047 |
| MoE-POT-S (12 Datasets) | 0.0533 | 0.0056 | 0.0062 | 0.0032 | 0.0383 | 0.0043 |

Table 15: Zero-shot L2RE on the six base evaluation datasets after pre-training on increasingly heterogeneous data mixtures (6, 10, and 12 datasets). Lower values are better.

The zero-shot L2 Relative Error (L2RE) for both models across the different pre-training configurations is presented in Table 15.

**DPOT-S (Dense Model)**   The performance of DPOT-S is unstable and often degrades as more heterogeneous data is introduced. While adding four datasets (from 6 to 10) yields minor improvements on some tasks, it causes significant performance degradation on others (e.g., DR and CFDBench). Expanding to 12 datasets results in a notable performance collapse on several tasks (e.g., NS(1e-3), SWE, DR) compared to the original 6-dataset training. This confirms that the dense architecture suffers from negative transfer, where the model's capacity is overwhelmed by conflicting information from diverse physical systems.

**MoE-POT-S (Sparse Model)**   In stark contrast, MoE-POT-S demonstrates remarkable robustness. As the number of pre-training datasets increases from 6 to 12, its performance remains stable or even improves on most tasks (e.g., CNS and CFDBench). The slight variations in error are minor compared to the drastic fluctuations observed with DPOT-S. This stability indicates that the MoE architecture effectively mitigates negative transfer by allowing different experts to specialize in distinct physical dynamics, thereby preventing knowledge conflict.

## C.6 Extended Interpretability Analysis: Emergence and Generalization of Router Specialization

This appendix expands on the interpretability analysis in Section 5.4, which demonstrated the router-gating network's ability to classify input data by its source PDE. This capability is not explicitly

programmed; the router only processes tokenized inputs, and the model's loss function is based on prediction error, not a classification objective. Here, we investigate two key questions: (1) How does this specialization emerge during pre-training? (2) Does this learned capability generalize to entirely new, OOD datasets?

**Emergence of Specialization During Training**    We first examine how the router's classification ability develops. We tracked the dataset classification accuracy at different stages of pre-training, using the method described in Section 5.4. The results, shown in Table 16, confirm that this is an emergent property. Initially (Epoch 50), the accuracy is low, but it rapidly improves and reaches 100% by Epoch 250. This demonstrates that the router learns to distinguish between data distributions as part of the end-to-end optimization process.

| Epoch | NS(1e-5) Accuracy | CFDBench Accuracy |
|---|---|---|
| 50 | 2% | 70% |
| 150 | 80% | 78% |
| 250 | 100% | 100% |

Table 16: Evolution of the router's dataset classification accuracy over training epochs. The accuracy steadily improves, eventually reaching 100% as the experts and router co-specialize.

**Generalization to Out-of-Distribution Datasets**    To test if this specialization generalizes beyond the pre-training distributions, we evaluated the router's classification performance on two OOD tasks from the Poseidon benchmark [20]: Wave-Layer and Wave-Gauss. These datasets represent novel physical systems unseen during pre-training.

| Unseen Dataset | Block-1 Accuracy | Block-2 Accuracy |
|---|---|---|
| Wave-Layer | 100% | 100% |
| Wave-Gauss | 100% | 100% |

Table 17: Router classification accuracy on unseen (OOD) downstream tasks. The perfect accuracy demonstrates strong generalization.

As shown in Table 17, the router achieves 100% classification accuracy on both unseen tasks. For instance, when processing the Wave-Layer dataset, the router in Block-2 consistently activated a sparse subset of experts (e.g., Expert 11 at 100% usage, Expert 1 at 79%), with other experts receiving minimal or zero activation.

Taken together, these results provide strong evidence that the router's ability to identify PDE types is an emergent property of joint optimization. To minimize the global prediction loss across heterogeneous datasets, the model learns to partition its knowledge, routing inputs with similar dynamics to specialized experts. This process effectively mitigates the negative transfer that hinders dense architectures. Crucially, the perfect classification accuracy on OOD data demonstrates that the router is not merely memorizing training distributions. Instead, it learns to recognize fundamental properties of the underlying physics from the tokenized solution data, a capability that generalizes to novel systems.

