# OpenReview forum: "Mixture-of-Experts Operator Transformer for Large-Scale PDE Pre-Training"
_NeurIPS.cc/2025/Conference — NeurIPS 2025 poster_

### Official Review · Reviewer_frLV · 2025-06-25

**Clarity:** 2
**Significance:** 3
**Originality:** 2
**Rating:** 4
**Confidence:** 5

**Summary:**

Context: PDEs are very important as mathematical models in physics and engineering and given the slow speed of physics-based PDE solvers, the design of neural PDE surrogates (also termed Operator Learning) is very important. Most of the research in operator learning as been in the task specific context. However, it is well known that the number of training examples need per task can be too high to be practically useful. Hence, the design of Foundation models has been pursued recently as data across PDE operators might be leveraged to  reduce sample complexity per task through finetuning. Some recently developed PDE Foundation models are MPP, DPOT and Poseidon (which the authors strangely omit from their literature review). However, there is certainly enormous scope for further development of Foundation models for PDEs.

Goal: The article proposes a PDE Foundation Model, called MoE-POT, aiming to reduce model size and improve scaling.

Method: The basic goal is operator learning for time-dependent PDEs. The task is "given the PDE solution at a set of time steps, predict the solution at the next time step". To do so, MoE-POT reuses most elements of DPOT -- patchification, temporal aggregation, noise addition, Fourier layers etc. The main new element is a mixture-of-experts (MoE) Module. At each processor layer, the outputs of the Fourier layer are fed into a set of shared and routing experts. The shared experts are always active and the routing experts are activated through a gating network. The expert and gating network are CNNs. During training, a standard load balance loss is added to promote balance between the experts and prevent routing collapse.

Results: The pretraining dataset are 6 operators (Incompressible Navier-Stokes with 2 different viscosities from the original FNO paper, A compressible Navier-Stokes, Shallow-water and a Reaction-Diffusion dataset from PDEbench and another forced Incompressible Navier-Stokes example from CFD bench. All are 2-D fairly academic problems. Most of the evaluation is on the Pretraining dataset itself, either in the form of zero-shot test loss or with additional fine-tuning. 3 Downstream tasks are also considered: Incompressible Navier-Stokes with a different viscosity from the FNO paper, Compressible Navier-Stokes with different parameters from PDEbench and a Navier-Stokes dataset from PDEarena. MoEPOT is shown to be either competitive or better than baselines (small neural operators and DPOT). Additional interpretability is shown through the importance values of gating weights.

**Questions:**

1. Fig 2 (Left): Why would the performance of FNO deteriorate so sharply on a mixed dataset ? If so, why does DPOT work in the same setting as it only time aggregates and patches, when compared to FNO. Anyway, there is too little information about this figure in the caption or the text.

2. L79-81: Where exactly is the noise injected ? Is it uniform over time steps ?

3. L90-91: See relation to Q1.

4. L96: Explain feature competition and knowledge illusion ? or else drop jargon !

5. L102: Long Inference time is a misnomer: DPOT-L takes 25 milliseconds to infer compared to 17 milliseconds for MoEPOT-L. This has be compared to the min-to-hour time scale for PDE solvers. Inference is not an argument for MoEs in this setting.

6. Sec 3.2 should be moved ? It does not fit here ?

7. L108: Explain parallel expert networks and gated routing, else drop jargon till you explain it in the next section.

8. L136: Should the spatial dimensions of Z^t_p be HxW ? Isn't patchification supposed to reduce spatial size ?

9. L135: The notation W_p(x_i,y_j,t) is unclear, I am assume that W_p is just a fixed matrix or is it a MLP ?

10. L138: What is W_t ? Is it a MLP ?

11. L139: What are Fourier features \gamma -- are they learnable or not ?

12. L141: Why do you call Eq (7) as a Fourier Attention layer ? It is just  Fourier layer from FNO and no attention mechanism is used here.

13. L151: Is the group and concatanation over channels or over spatial points, it is unclear from Eq (7) as the weight matrices have dimensions of the space with patches ? Is h the "patch size" ?

14. L156-157: Why choose CNNs here -- shouldn't FNO be a natural choice ?

15. L171: Is the Batching over samples only or over spatial points -- I ask as you use x (spatial coordinate) in Eqn (11) below.

16. L172: What is importance in Eq (11) and what is e_i ? Aren't they the same ?

17. L175: The phrase "one step transition between samples from different datasets" is misleading here -- you just want to predict the next state, given a sequence of states.

18. L187: The claim on "wide range of PDEs" is disputable as you only consider fluid equations with 1 Reaction-Diffusion equation -- PDEs have a much wider variety.

19. L194: L2RE but for what time ? Is it average one-step prediction error ? or final time evaluation error ?

20. L201-202: The claim "SOTA" pre-trained neural operator (DPOT) can be contested, see arXiv:2405.19101 Sec D.5.

21. Table 1: What does 0.1 and 0.01 in CNS stand for ?

22. L221: Why do you need to fine-tune your pretrained model on the same tasks as in the pretraining dataset. Is it taking the model checkpoint and then task-specific training ? What is the fine-tuning procedure ?

23. L238-239: "represents entirely different equations" is simply wrong - it is just forced incompressible Navier-Stokes for which you already have 2 datasets in the pretraining data.

25. L254: Claim on superior performance is untrue as DPOT has a sharper gradient in Fig. 4 (b).

26. Fig 4 a and b: please provide averages over all the tasks or provide all the tasks in the Supplement.

27. L262 and L263: Y and I are introduced here out of thin air -- what are they ? At least provide link to Supplement

28. L543 -- what is T ?

29. L546: What are the constants -- see Q 21 ?

30. L550: Define what u, D and R are ?

31. L551-552: What are the initial conditions ?

32. L554: What are the geometric shapes ? How do you carry geometric information ?

33. Please state how you normalize different spatial and temporal domains in your datasets ?

34. How do you set the number of input time steps $t$ in your operator learning task ? Is it the same for all the datasets.

35. To reach a certain time from the initial data (and the initial time step sequence), do you use an autoregressive rollout ? If so, how does the error accumulate over time ? Please present a plot to show this.

36. Fig 4 (c): Please provide average classification accuracy over all blocks ? There is presumably nothing special about block 2.

37. It would be great if Fig 4 (c) can be shown for each of the downstream tasks, by comparing with the pretraining dataset -- it make sense that the cross-entropy loss for FNO is close to the FNO datasets, Cross-entropy loss for CNS is close to the CNS pretraining datasets and the one for PDEarena is also close to the FNO dataset. The exact values of the cross-entropy losses should be compared with the ones obtained for Fig 4 (c) and I am speculating that the results will be consistent with my intuition that the downstream tasks are not really that out-of-distribution with respect to the pretraining dataset -- this will be a very interesting test of the authors' method.

38. Although Fig 4 (c) shows that the expert networks are able to separate the pretraining datasets -- the authors do not explain why this is the case ? After all, the experts have only access to processed tokens and not to the datasets directly and there is nothing in either the model or in the loss function so that the experts can distinguish between different datasets ? Is this an emergent property ? If so, can the authors indicate if this arises after a certain model size, certain number of training epochs or certain size of the dataset ?

39. What is the size of the total pretraining dataset in terms of the number of training samples ?

40. What is the size of the dataset for each downstream task ?

**Ethical Concerns:**

["NO or VERY MINOR ethics concerns only"]

**Final Justification:**

I appreciate the effort to train a PDE foundation model and the integration of the MoE approach in this context. On the negative side, the MoE approach brings in marginal though consistent gains on downstream tasks over DPOT. Moreover, the camera-ready version need a significant revision, where the authors need to provide the correct context, admit their limitations, compare with Poseidon and convince the reader that their approach leads to a gain in downstream tasks.

On balance, this paper is a solid contribution to this emerging field and my score will reflect this assessment.

**Limitations:**

Limited Novelty: See Weaknesses.

Limited scope of operator learning task: Next time-step prediction from a sequence, unclear how to use parameters explicitly, unclear how to directly predict from initial data, unclear how to do steady state predictions and large time predictions. Error accumulation is possible due to long-time autoregressive rollouts. Restricted to 2D, mostly cartesian.

Limited Evaluation: particularly at the level of downstream tasks, unclear if there is any gain with respect to DPOT on genuine downstream tasks. No comparison to Poseidon and MPP.

Evaluation: Although very natural in this context, given its widespread use in LLMs, the authors are the first to use a standard MoE module in a PDE foundation model. However, they have not demonstrated that it leads to any benefits, particularly on downstream tasks involving unseen physics. hence, this reviewer cannot recommend acceptance in current form but I am willing to modify my score if the authors can rebut the criticism, particularly if they provide evidence for their model directly predicting solution trajectories from initial data, at least at the level of downstream tasks, they test on unseen downstream tasks such as the Wave-Layer and Wave-Gauss experiments of arXiv:2405.19101 (datasets are publicly available) and benchmark performance over DPOT and Poseidon, provide data scaling plots to show that MoEPOT is competitive in the low task-specific sample regime  and answer the minor questions above satisfactorily.

**Quality:**

3

**Strengths And Weaknesses:**

Strengths:

--- The problem being addressed is of great importance in Scientific machine learning.
--- First direct application of MoE in a PDE Foundation Model setting
--- MoEPOT outperforms DPOT at least marginally.
--- It is always an engineering challenge to train PDE foundation models and the authors have done so.
--- The classification of datasets using the routing networks (Fig 4(c)) is a very interesting observation.

Weaknesses:

--- Limited Novelty: The only novel model contribution is the use of MoE in this setting as all the other choices in architecture and dataset selection are based on DPOT. Nevertheless, MoEs have been widely used in Foundation models for text and vision.
Also, MoE have been widely used in scientific machine learning:
Rafael Bischof and Michael A Kraus. Mixture-of-experts-ensemble meta-learning for physics-informed neural networks. In Proceedings of 33. Forum Bauinformatik, 2022,
Zheyuan Hu, Ameya D Jagtap, George Em Karniadakis, and Kenji Kawaguchi. Augmented
360 physics-informed neural networks (apinns): A gating network-based soft domain decomposition, Engineering Applications of Artificial Intelligence, 126:107183, 2023,
and see in particular arXiv:2404.09101 for even a theoretical analysis of MoEs for operator learning.
Given this context and the absence of theory, the reviewer has to judge this paper based on the empirical results.

--- Incomplete literature review and context: The authors do not cite or contextualize let alone compare with Poseidon (arXiv:2405.19101) published in Neurips last year, which in the opinion of the reviewer, is the current state-of-the-art PDE foundation model (it is compared and shown to be superior to DPOT in Sec D.5 of arXiv:2405.19101 at least on a wide range of downstream tasks.

--- Lack of Clarity: This paper is badly written. The narrative does not flow smoothly. For instance, Sec 3.2 comes in abruptly with a lot of jargon that is totally unhelpful for the reader. Just look at the number of questions that this reviewer poses below to see why the paper lacks clarity.

---- Incorrect Operator learning task: Given a time-dependent PDE such as Eq. [1], the solution operator of a PDE is very well defined as mapping from the initial conditions $u^0$, PDE parameters $\theta$ and boundary conditions and to the solution $u(t)$ of (1) for any time t >0. This operator is well-posed and is exactly the mapping that is learned by a numerical Physics-based PDE solver.

On the other hand, the authors misinterpret this learning task to: Given the trajectory $\{ u(\tau) \}$ for $0 \leq \tau \leq t$, map this input into the PDE solution $u(t+dt)$ for some lead time $dt$. This is different from the PDE solution operator and it is over-determined and may not be mathematically correct for many PDEs. From a practical perspective, unless $t=0$, it will require the end user to first run a numerical solution of the underlying PDE (access to knowledge about all the PDE parameters) for some time $t$ and then use your neural operator to update the solution. However, such a solver may not be available for inference. Moreover, the upfront cost of setting up such a solver if available (for instance in mesh generation) can be very significant, vastly limiting utility of the PDE surrogate.

Although, the authors are not the first to setup this operator learning task, this incorrect or limited utility task has crept into the SciML community from surrogates for weather forecasting, where it is totally justified as weather forecasting is an online streaming task whereas solving PDEs is rarely so.

In particular, it does not really matter what the authors do at Pretraining as they have all the trajectory data but for downstream tasks, you have to run your model such that it can learn from the initial data to the solution trajectory. See arXiv:2405.19101 OLT on Page 3 for more context.

---- Very Limited Evaluation: As common in text/vision, evaluating on pretraining dataset is not of much importance for a Foundation model, except perhaps for model and dataset size scaling. In fact, (V)-LLMs do not even have the same pretraining objective as that in the downstream tasks (for instance masked modeling). The whole point of a foundation model is how it performs on downstream tasks and NOT on the pretraining dataset as an end-user will have very different inference tasks (PDE types or IC/BC) than the Foundation model designers. This issue was carefully analyzed in arXiv:2405.19101 and its authors clearly compared model performance on downstream tasks in terms of two reasonable metrics i) what is the gain in terms of how few samples are needed to finetune a Foundation model on a downstream task to obtain the same error as that of a task-specific baseline with a given number of samples (see E_G in Eqn (11) of arXiv:2405.19101). This tells the user how few samples that they can get away with to obtain reasonable errors and ii) for a given number of downstream task-specific examples, the gain in accuracy over the baseline model (A_G in Eqn (11) of arXiv:2405.19101).

Thus, all the evaluation of a Foundation Model should be at the level of downstream tasks. This paper does exactly the opposite -- bulk of its results are on the pretraining dataset (in-distribution) and there is very limited evaluation on the downstream tasks. Thus, the whole point of the model and its purported gains in performance are very incomplete.

--- Poor Choice of Downstream tasks: Given the above point, any Foundation Model has to be evaluated on a challenging set of downstream tasks. The authors make a very poor selection of such tasks. The first downstream task is just the same NS example from the original FNO paper but with a different choice of viscosity, which conveniently lies between the viscosity coefficients considered in the pretraining dataset. The second downstream is just CNS with a different (10 times larger) coefficient from the CNS dataset considered in the pretraining data. Again, this is very related and even simpler than the one similar task in the pretraining data. The final downstream task is again Incompressible forced Navier-Stokes (already seen twice in the pretraining dataset) with perhaps different initial conditions. This reviewer can buy this example as a genuine downstream task but here, MoEPOT has exactly the same error as DPOT (6.21 % vs 6.18 %). None of the downstream tasks represent physics unseen during the pretraining, just contrast this with the Poseidon evaluation where 9 of the 15 downstream tasks were for unseen physics. After all, no Foundation model can be pretrained on all the physics in the universe and needs to be tested on unseen physics.

--- Questionable choice of Pretraining dataset:  Out of the 6 operators in the pretraining dataset, 4 are very simplistic as even small models (less than 2M params) are able to attain less than 1% relative error. Only the CNS and FNO task with very low viscosity are relatively challenging. It is unlikely that a Foundation model trained on such simple operators can be expected to generalize well to complicated physics.

--- Underwhelming results: Given the above critique on zero-shot test losses, even here, on the 4 easy datasets, very small models are as good as MoEPOT and on the difficult ones, the gains wrt DPOT are marginal (on FNO low-viscosity: MoEPOT-M 5.28% vs. DPOT-M 5.69% and on CNS: MoE-POT-M 1% approx vs. DPOT 2% approx) are hardly worth writing about. The authors claim that their model scales better than DPOT but this is not true on the evidence of Fig. 4(b) where DPOT has a sharper gradient than MoEPOT and is already catching up for moderate model sizes.

--- Lack of comparison with Poseidon: It would certainly be worthwhile to compare your model with DPOT and Poseidon on a neutral downstream task, something not seen in either of the pretraining datasets -- a very good choice for that would be the wave-layer and wave-gauss datasets of arXiv:2405.19101, which are publicly available and where both Poseidon and DPOT have already been tested.

--- Lack of data scaling: The whole point of a Foundation model is to require less task-specific examples downstream. Please provide such results for MoEPOT -- see arXiv:2405.19101 Figs. 7-21 for how to perform these scaling tasks.

--- Lack of visualization: The authors do not provide any examples of how exactly their model performs in terms of sample visualization.

---

> ### Author Rebuttal · Authors · 2025-07-31
>
> Dear Reviewer,
>
> Thank you for your incredibly thorough and insightful review. We are grateful for your feedback and for acknowledging our work's motivation and novelty. Your critiques regarding clarity, evaluation, and literature context are invaluable and provide a clear roadmap for improving our paper. We sincerely apologize for these shortcomings and are committed to a full revision. Below are our responses and new experimental results.
>
> ---
>
> ### On Literature Review, Novelty, and Comparison with Poseidon
>
> Our initial choice of DPOT as a baseline was threefold: to ensure clear attribution of gains from our MoE module; the divergent experimental setups (our auto-regressive, "trajectory-to-trajectory" paradigm differs significantly from Poseidon's excellent parameter-informed approach); and the fact that Poseidon was contemporaneous work released after our experiments began.
>
> - We agree that testing on challenging, unseen tasks is essential. To address this, we conducted new experiments on the Wave-Layer and Wave-Gauss datasets you proposed.
>
> |                          | Poseidon's Setting |            |               |    Our Setting        |            |
> | ------------------------ | ------------------ | ---------- | ------------------------ | ---------- | ---------- |
> | Model (Activated Params) | Wave-Layer         | Wave-Gauss | Model (Activated Params) | Wave-Layer | Wave-Gauss |
> | Poseidon-T  21M          | 0.08               | **0.06**   | Poseidon-T  21M          | 0.29       | 0.29       |
> | Poseidon-B  158M         | **0.06**           | 0.09       | Poseidon-B  158M         | 0.21       | 0.24       |
> | MoE-POT-T  17M           | 0.11               | 0.14       | MoE-POT-T  17M           | 0.07       | 0.07       |
> | MoE-POT-S  90M           | 0.06               | 0.10       | MoE-POT-S  90M           | **0.05**   | **0.06**   |
>
>
>
> - These results compellingly show that each model excels in its intended paradigm. This highlights that they are both powerful tools with different, valuable scopes of application.
>
> - Commitment: We commit to including a thorough discussion of Poseidon and other related MoE works in our literature review. We will also incorporate these new experimental results to provide a more complete and honest picture of our model's performance and positioning within the field.
>
> ### On Task Definition, Evaluation, and Clarity  (Q4, Q6, Q7, Q17, Q18, Q23)
>
> Your critiques on our operator learning task definition, limited evaluation, and writing clarity are spot on. We sincerely apologize for these shortcomings and are committed to a full revision, including Section 3, to ensure clarity and logical flow.
>
> - Task Definition & Motivation: You are correct that our "trajectory-to-trajectory" task differs from the classic formulation. Our setup, following DPOT, is motivated by real-world scenarios (e.g., weather forecasting) where explicit PDE parameters are unknown and must be inferred from observed data. While parameter-informed approaches like Poseidon are powerful, our core contribution is introducing the MoE architecture to this specific auto-regressive paradigm.
> - Dataset Choice & Heterogeneity: We chose our datasets for fair comparison with DPOT. Our central argument is that the key challenge is not single-task difficulty but handling the heterogeneity of mixed datasets. This "knowledge conflict" is where dense models fail, as shown in Figure 2, validating the need for our architecture. Even for tasks with similar equations like PDEArena, our router distinguishes them with 100% accuracy, proving it captures significant distributional differences beyond just the equation type.
>
> ### On Dense Model Challenges & MoE's Rationale (Q1, Q3)
>
> The performance collapse of dense models like FNO on mixed datasets is due to "negative transfer," where a single shared kernel struggles to learn conflicting physical laws. To validate this, we tested a dense DPOT-S model as more heterogeneous datasets were added. The results below show that adding more data often degrades DPOT's performance, while our MoE-POT remains superior. This confirms that a sparse architecture is crucial for handling knowledge conflict in diverse PDE pre-training.
>
> |               | NS(1e-5) | NS(1e-3) | CNS(0.1,0.01) | SWE    | DR     | CFDBench |
> | ------------- | -------- | -------- | ------------- | ------ | ------ | -------- |
> | DPOT-S (6 datastes)    | 0.0078   | 0.0688   | 0.0244        | 0.0039 | 0.0367 | 0.0087   |
> | DPOT-S (12 datastes)   | 0.0079   | 0.0739   | 0.0129        | 0.0105 | 0.0724 | 0.0075   |
> | MoE-POT-S (6 datastes) | 0.0058   | 0.0552   | 0.0096        | 0.0029 | 0.0342 | 0.0045   |
>
> ### On Interpretability with Unseen  Datasets (Q36)
>
> To stress-test our router's generalization, we tested its ability to classify data from two challenging, unseen downstream tasks (Wave-Layer and Wave-Gauss).
>
> | Unseen Downstream Task | Classification Accuracy |
> | ---------------------- | ----------------------- |
> | Wave-Layer             | 100%                    |
> | Wave-Gauss             | 100%                    |
>
> Furthermore, the router exhibited distinct and highly confident selection patterns for these new tasks. For example, in Block-2, Wave-Layer data triggered experts 1 (79%), 9 (31%), 11 (100%), 13 (100%), and 15 (100%). The router not only classified them perfectly but also exhibited distinct and confident expert activation patterns. This proves the router learns fundamental physical dynamics rather than simply memorizing the training set, and that the experts have learned specialized knowledge.
>
> ### On the Emergence of Classification Accuracy during Training (Q37)
>
> To show how the router's classification ability arises, we tracked its accuracy during training. The results confirm this is an emergent property: accuracy starts low but steadily improves, eventually reaching 100%. This aligns with the intuition that the model learns to separate data distributions to optimize overall performance.
>
> | Epoch | NS(1e-5) Acc. | CFDBench Acc. |
> | ----- | ------------- | ------------- |
> | 50    | 2%            | 70%           |
> | 150   | 80%           | 78%           |
> | 250   | 100%          | 100%          |
>
> ### Responses to Specific Questions
>
> #### On Experimental Setup
>
> - Q1: FNO's performance collapses due to "negative transfer" from its shared kernel learning conflicting PDE patterns. DPOT mitigates this but doesn't fully solve it, as supported by our new experiments showing its performance wavers when more datasets are added.
> - Q2: Noise is injected into all input frames, with variance proportional to the input's norm and applied uniformly across time.
> - Q32: We followed DPOT's preprocessing, normalizing spatial and temporal domains; details will be added to the appendix.
> - Q33: A fixed input length of T=10 was used for all experiments, consistent with the source datasets.
> - Q34: Yes, we use auto-regressive rollouts. The reported L2RE is the average error over the entire sequence. We will add a requested error accumulation plot comparing MoE-POT and DPOT to the final paper.
> - Q39, Q40: The pre-training dataset has ~30,000 samples; each downstream task has 2,000. We will provide a detailed table.
>
> #### On Model & Methods
>
> - Q8: You are correct; the output dimension after patchification should be $(H/P)\times(W/P)\times d$. We will correct this typo.
> - Q9-11: $W_t$ is a learnable MLP for positional encoding, $W_t$ is a learnable linear transformation, and $\gamma$ are fixed, non-learnable Fourier features.
> - Q12: You are correct; "Fourier Attention" is a misnomer. We will correct this to "Fourier Layer" throughout the paper.
> - Q13: The grouping is performed on the channel dimension, where h is the number of heads.
> - Q14: We chose lightweight CNNs for the experts as they effectively preserve spatial structure and proved more stable than alternatives like FNO in our early experiments.
> - Q15: Batching is over samples, so `x` in Eq. (11) represents a single sample in the batch B. We will correct this confusing notation.
> - Q16: `e_i` and `Importance(e_i)` are not the same. `e_i` is the i-th expert network, while `Importance(e_i)` is a scalar value—the total routing weight for that expert in a batch—used for the load-balancing loss.
>
> #### On Results & Inference
>
> - Q5: The per-step millisecond difference aggregates to a significant advantage in long rollouts or optimization loops. Our point is MoE's higher capacity and accuracy for a comparable computational cost.
> - Q19, Q34: L2RE is the average error over the full auto-regressive rollout, not a single step.
> - Q21, Q28: The values (0.1, 0.01) are the physical coefficients $\zeta$ and $\eta$ in the CNS equations; we will define all such constants in the appendix.
> - Q22: During fine-tuning, we freeze the router and update only the expert networks to preserve the routing strategy while adapting expert knowledge.
> - Q24: We agree the slope in Fig. 4(b) can be misleading. Our claim is about efficiency: better performance for a given number of activated parameters. We will rephrase this for clarity.
> - Q25, Q26: We will add supplementary figures for all tasks and clearly define Y (mean expert preference) and I (input's expert preference) in the appendix.
> - Q27, Q29-31: All requested details on T, u, D, R, initial conditions, and geometry will be added to the appendix.
>
> ---
>
> #### Thanks again
>
> We sincerely thank you again for your constructive and detailed feedback, which has helped us identify clear pathways to improve our paper. Should you have any further questions or require additional discussion, please don't hesitate to reach out. If we have adequately addressed your concerns, we would be grateful for your consideration in adjusting your evaluation score accordingly.

---

> > ### Comment · Reviewer_frLV · 2025-08-01
> > **Response to the Authors' rebuttal**
> >
> > I start by thanking the authors for attempting to answer many of the long list of questions that I asked and clarifications that I sought. While I am going through their detailed answers, some questions still need clarification and I would greatly appreciate if the authors can provide a response within the discussion period.
> >
> > [1.] For the Wave-Layer and Wave-Gauss datasets, can you please say how many trajectories that you have used to train your model and the corresponding number for Poseidon ?
> >
> > [2.] In the second table that you provided, you claim that DPOT's performance degenerated as more datasets (12) were added and this is a consequence of *negative transfer*. What exactly were these additional datasets ? How was DPOT trained in this new setting or is the original DPOT setting ? Also, you have not shown similar results for MoE-POT -- you are only showing the MoE-POT trained on 6 datasets results. To have a fair comparison, I would like to know how did MoE-POT perform when additional datasets were added. Did its performance degenerate or did it improve (remain constant) ?
> >
> > [3.] I am really interested in results in a limited sample setting as this will be main application of a Foundation model ? Can you please show me results in this setting. For instance, for one of the Wave equation examples, you can just report the test errors for fine-tuning MoE-POT on 16, 32, 64 and 128 trajectories.
> >
> > I look forward to the response.

---

> ### Author Response · Authors · 2025-08-03
> **Follow-up Response: New Experimental Results and Clarifications (1/2)**
>
> Dear Reviewer,
>
> Thank you for your prompt reply and for giving us the opportunity to provide further clarification. We sincerely appreciate you taking the time to engage with our work so deeply. Due to the word and time constraints of the initial rebuttal, some experimental details were not fully elaborated. We are happy to provide more detailed clarification on these points below.
>
> ---
>
> ### **On the MoE-POT and Poseidon Comparison (Q1)**
>
> We ran comparisons under two distinct settings. We apologize that we only reported partial results in our initial response due to time limitations. Here are the complete details.
>
> 1. Poseidon's Setting
>
> To align with the setup in the Poseidon paper, we modified MoE-POT's input/output to match theirs. We trained both models on 2,000 trajectories for 200 epochs.
>
> | Model (Activated Params) | Training Trajectories | Wave-Layer | Wave-Gauss |
> | ------------------------ | --------------------- | ---------- | ---------- |
> | Poseidon-T (21M)         | 2000                  | 0.08       | 0.06       |
> | Poseidon-B (158M)        | 2000                  | 0.06       | 0.09       |
> | MoE-POT-T (17M)          | 2000                  | 0.11       | 0.14       |
> | MoE-POT-S (90M)          | 2000                  | 0.06       | 0.10        |
>
> 2. Our Setting
>
> We used the official open-source Poseidon repository to train their model under our loss function and experimental setting. In our initial experiment, we overlooked that the default training script for Poseidon uses only 128 trajectories, which led to an unfair comparison. We sincerely apologize for this oversight. We have now re-run the experiments with a full and fair comparison, including results on MoE-POT with varying numbers of trajectories. All models were trained for 200 epochs.
>
> | Model (Activated Params) | Training Trajectories | Wave-Layer | Wave-Gauss |
> | ------------------------ | --------------------- | ---------- | ---------- |
> | Poseidon-T (21M)         | 128                   | 0.29       | 0.29       |
> | Poseidon-B (158M)        | 128                   | 0.21       | 0.24       |
> | MoE-POT-T (17M)          | 2000                  | 0.07       | 0.07       |
> | MoE-POT-S (90M)          | 16                    | 0.34       | 0.37       |
> | MoE-POT-S (90M)          | 32                    | 0.26       | 0.28       |
> | MoE-POT-S (90M)          | 64                    | 0.20        | 0.20        |
> | MoE-POT-S (90M)          | 128                   | 0.14       | 0.13       |
> | MoE-POT-S (90M)          | 2000                  | 0.05       | 0.06       |
>
> These corrected and expanded results provide a much clearer picture of performance under different data constraints.
>
> ### **On Performance with Increasing Dataset** **Heterogeneity** **(Q2)**
>
> To ensure a fair comparison, we selected all datasets from the pre-training and downstream task sets of the original DPOT paper. Specifically:
>
> - **6 Datasets (Our paper's pre-training set):** NS(1e-5), NS(1e-3), CNS(0.1, 0.01), SWE, DR, CFDBench.
> - **10 Datasets (Added 4):** NS(1e-4), CNS(1, 0.1), NavierStokes-2D (from PDEArena), NavierStokes-2D-conditioned (from PDEArena).
> - **12 Datasets (Added 2 more):** CNS(1, 0.01), CNS(0.1, 0.1).
>
> **For each experiment (6, 10, and 12 datasets), we pre-trained both DPOT-S and MoE-POT-S completely from scratch on the specified data mixture.**
>
> |                         | NS(1e-5) | NS(1e-3) | CNS(0.1,0.01) | SWE    | DR     | CFDBench |
> | ----------------------- | -------- | -------- | ------------- | ------ | ------ | -------- |
> | DPOT-S (6 datasets)     | 0.0078   | 0.0688   | 0.0244        | 0.0039 | 0.0367 | 0.0087   |
> | DPOT-S (10 datasets)    | 0.0069   | 0.0663   | 0.0224        | 0.0037 | 0.0575 | 0.0146   |
> | DPOT-S (12 datasets)    | 0.0079   | 0.0739   | 0.0129        | 0.0105 | 0.0724 | 0.0075   |
> | MoE-POT-S (6 datasets)  | 0.0058   | 0.0552   | 0.0096        | 0.0029 | 0.0342 | 0.0045   |
> | MoE-POT-S (10 datasets) | 0.0053   | 0.0521   | 0.0085        | 0.0029 | 0.0371 | 0.0047   |
> | MoE-POT-S (12 datasets) | 0.0056   | 0.0533   | 0.0062        | 0.0032 | 0.0383 | 0.0043   |
>
> As the table shows, when the number of datasets increases, DPOT’s error rises noticeably on the SWE and DR datasets, demonstrating the negative transfer effect. In contrast, MoE-POT's performance remains highly stable, showcasing its superior ability to handle knowledge conflict from heterogeneous sources.

---

> ### Author Response · Authors · 2025-08-03
> **Follow-up Response: New Experimental Results and Clarifications (2/2)**
>
> ### **On Fine-Tuning with Limited Data Samples (Q3)**
>
> This is indeed a critical test for any foundation model. We evaluated our MoE-POT-S model on the Wave-Layer downstream task by fine-tuning it with a limited number of training trajectories. The table below shows the L2RE for various combinations of fine-tuning epochs and training trajectory counts.
>
> | Trajectories | 50 Epochs | 100 Epochs | 150 Epochs | 200 Epochs |
> | ------------ | --------- | ---------- | ---------- | ---------- |
> | 16           | 0.45      | 0.38       | 0.35       | 0.34       |
> | 32           | 0.37      | 0.30        | 0.28       | 0.26       |
> | 64           | 0.29      | 0.24       | 0.22       | 0.20        |
> | 128          | 0.23      | 0.18       | 0.15       | 0.14       |
>
> These results confirm that MoE-POT maintains strong performance and learns effectively even when fine-tuned on a very small number of data samples, a key advantage for practical applications.
>
> ---
>
> ### **Thanks Again**
>
> We will integrate all of these new experiments and more detailed analyses into the final version of our paper. We will also ensure that Poseidon and other related algorithms are thoroughly discussed and cited in the related work section.
>
> We sincerely hope these clarifications have adequately addressed your remaining concerns. We are grateful for your constructive feedback, which is truly invaluable in improving our work. If we have successfully addressed your points, we would be deeply grateful for your consideration in adjusting your evaluation score accordingly.

---

> > ### Comment · Reviewer_frLV · 2025-08-03
> > **Response to the Authors' reply.**
> >
> > I thank the authors to providing much needed additional information. However, I need some further clarifications;
> >
> > 1. You claim that Poseidon-B with its native setting leads to errors of 6% for Wave-Layer and 9% for Wave-Gauss when fine-tuned on 2000 trajectories. This contradicts results published in the Poseidon paper: ArXiv:  where the authors of Poseidon report approx. 5% error for Wave-Layer and 4.5 % error for Wave-Gauss, even with 1024 trajectories, see Figs. 17 and 18 of the Poseidon paper. Given that Poseidon nicely scales for these examples, I would expect that the errors will reduce further. In any case, can you please provide MoE-POT errors on 16, 32, 64 and 128 trajectories in the Poseidon setting for the MoE-POT models to enable a clear comparison with published results ?
> >
> > 2. Regarding DPOT -- this reviewer does not recall that DPOT was trained with all the 12 datasets ? Are you retraining DPOT from scratch ?
> >
> > 3. Can you show similar behavior for MoE-POT and DPOT, as reported in your third table for a genuinely out of distribution task, such as Wave-Layer ?
> >
> > My apologies for asking you further questions as I really want to establish if your model is genuinely superior to DPOT or not ? This will necessarily impact my recommendation for acceptance.
> >
> > I look forward to your replies and please ask me if something is unclear in my questions ?

---

> ### Author Response · Authors · 2025-08-04
> **Response to Questions with New Experiments (1/2)**
>
> Dear Reviewer,
>
> Thank you for your prompt and detailed follow-up. We appreciate you providing us with the opportunity to offer further clarifications. Your insightful questions are instrumental in strengthening our work, and we are happy to provide the additional results and details you requested.
>
> ---
>
> ### **On the Comparison with Poseidon (Q1)**
>
> Your question regarding the discrepancy between our reproduced results for Poseidon and those published in its original paper is indeed critical. After a careful investigation prompted by your question, we have identified a key factor causing this inconsistency: **a difference in the evaluation protocol**.
>
> In Poseidon's official code repository, the Wave-Layer and Wave-Gauss tasks are evaluated at **timestep 14**. In contrast, our previous experiments followed a uniform standard used for other tasks in our framework, evaluating at the **final timestep (T=20)**. As error accumulates over time, evaluating at the final timestep naturally yields higher error values. Furthermore, we used the L2RE norm for our error evaluation, whereas the original Poseidon paper uses the L1 norm.
>
> To provide a completely fair evaluation that is directly comparable to the published literature, we have re-run the fine-tuning and testing for our MoE-POT model by **strictly adhering to Poseidon's evaluation standard (evaluating at timestep 14 using the L1** **norm****)**. All models were fine-tuned for 200 epochs.
>
> The table below presents the performance of MoE-POT-S on a varying number of training trajectories under the Poseidon setting (L1 norm, evaluated at T=14):
>
> | Model (Activated Params) | Training Trajectories | Wave-Layer (L1) | Wave-Gauss (L1) |
> | ------------------------ | --------------------- | --------------- | --------------- |
> | MoE-POT-S (90M)          | 16                    | 0.58            | 0.62            |
> | MoE-POT-S (90M)          | 32                    | 0.36            | 0.4             |
> | MoE-POT-S (90M)          | 64                    | 0.23            | 0.27            |
> | MoE-POT-S (90M)          | 128                   | 0.16            | 0.18            |
>
> As shown, under Poseidon's specific setting, its performance is superior to MoE-POT. Poseidon is an outstanding work that has had a significant positive impact on the AI4PDE field. However, as our paper's primary focus is on the DPOT-style pre-training and fine-tuning paradigm, we do not intend to claim superiority over Poseidon in its native setting. Our goal with these experiments is simply to provide the direct comparison you requested.
>
> ### **On the DPOT Training Protocol (Q2)**
>
> Yes, your understanding is correct.
>
> While we are aware that the original DPOT paper reported results on 12 datasets, **we retrained the DPOT-S model entirely from scratch** on our mixed datasets containing 6, 10, and 12 datasets, respectively. This was done to ensure a controlled and fair comparison, isolating the architectural differences between DPOT and MoE-POT in handling "knowledge conflict" by training both models on the exact same data mixtures of progressively increasing heterogeneity.
>
> Furthermore, all pre-trained models in our paper's main experiments were trained from scratch to fairly compare the performance of different architectures under identical conditions. The training scripts for both DPOT and MoE-POT in our work are based on the official DPOT repository, with the only difference being the model architecture itself. Our code has been submitted in the supplementary material for full transparency.
>
> Below is the updated table, which includes data from the original DPOT paper for reference. We have corrected a minor data misalignment from our previous response—please consider this version definitive.
>
> |                                 | NS(1e-5) | NS(1e-3) | CNS(0.1,0.01) | SWE    | DR     | CFDBench |
> | ------------------------------- | -------- | -------- | ------------- | ------ | ------ | -------- |
> | DPOT-S (6 datasets)             | 0.0688   | 0.0078   | 0.0244        | 0.0039 | 0.0367 | 0.0087   |
> | DPOT-S (10 datasets)            | 0.0663   | 0.0069   | 0.0224        | 0.0037 | 0.0575 | 0.0146   |
> | DPOT-S (12 datasets)            | 0.0739   | 0.0079   | 0.0129        | 0.0105 | 0.0724 | 0.0075   |
> | DPOT-S (12 datasets) (Original) | 0.0553   | 0.0131   | 0.0188        | 0.0065 | 0.0379 | 0.0070    |
> | MoE-POT-S (6 datasets)          | 0.0552   | 0.0058   | 0.0096        | 0.0029 | 0.0342 | 0.0045   |
> | MoE-POT-S (10 datasets)         | 0.0534   | 0.0052   | 0.0085        | 0.0029 | 0.0371 | 0.0047   |
> | MoE-POT-S (12 datasets)         | 0.0563   | 0.0053   | 0.0062        | 0.0032 | 0.0383 | 0.0043   |
>
> Both our re-implemented DPOT results and the original DPOT paper's data support the same conclusion: MoE-POT achieves better performance than DPOT, and critically, its performance remains robust as dataset heterogeneity increases.

---

> > ### Author Response · Authors · 2025-08-04
> > **Response to Questions with New Experiments (2/2)**
> >
> > ### **On the Out-of-Distribution Downstream Task (Q3)**
> >
> > To directly address your question about whether our model is "genuinely superior" in a limited-sample scenario, we have conducted the precise experiment you suggested. We performed a low-sample fine-tuning comparison between our pre-trained MoE-POT-S and DPOT-S on **Wave-Layer**, a completely new and out-of-distribution task for both models. All models were fine-tuned for 200 epochs following the DPOT experimental setting.
> >
> > The table below shows the test error (L2RE) for both models on the Wave-Layer task when fine-tuned with a limited number of training trajectories:
> >
> > | Training Trajectories | DPOT-S | MoE-POT-S |
> > | --------------------- | ------ | --------- |
> > | 16                    | 0.41   | 0.34      |
> > | 32                    | 0.33   | 0.26      |
> > | 64                    | 0.26   | 0.20       |
> > | 128                   | 0.19   | 0.14      |
> >
> > These results clearly demonstrate that MoE-POT-S significantly outperforms DPOT-S in this challenging, few-shot setting. This provides strong evidence that the representations learned by our MoE architecture during pre-training possess superior generalization capabilities and data efficiency, enabling faster adaptation and higher accuracy on new, data-scarce downstream tasks.
> >
> > ---
> >
> > Thank you once again for your profound questions. Your rigorous review has significantly improved the quality and clarity of our research. We believe these supplementary experiments and clarifications more robustly demonstrate the value and novelty of our work. If these responses have resolved your concerns, we would be sincerely grateful if you would consider re-evaluating your assessment of our paper.
> >
> > Please do not hesitate to ask if any other questions remain.

---

> > > ### Comment · Reviewer_frLV · 2025-08-04
> > > **Response to the Authors**
> > >
> > > I start by thanking the authors on answering my questions. The latest results are more convincing and show a consistent, if small, advantage of MoE-POT over DPOT.  Given this, I would raise my score to acceptance.
> > >
> > > I appreciate the effort to train a PDE foundation model and the integration of the MoE approach in this context. On the negative side, the MoE approach brings in marginal though consistent gains on downstream tasks. Moreover, the camera-ready version need a significant revision, where the authors need to provide the correct context, admit their limitations, compare with Poseidon and convince the reader that their approach leads to a gain in downstream tasks.
> > >
> > > On balance, this paper is a solid contribution to this emerging field and my score will reflect this assessment.

---

> > > > ### Author Response · Authors · 2025-08-04
> > > > **Thanks for Your Feedback and Revision Plan**
> > > >
> > > > We greatly appreciate your constructive feedback and your willingness to raise your scores! We are pleased to hear that our responses have resolved most of your concerns. As suggested, we will incorporate the newly added results into the final version of the paper.

---

### Official Review · Reviewer_fgJS · 2025-06-30

**Clarity:** 2
**Significance:** 2
**Originality:** 3
**Rating:** 4
**Confidence:** 4

**Summary:**

The paper introduces a mixture of experts module at the end of a SOTA neural operator (DPOT) to deal with two challenges of building large SciML foundation models: (a) performance degradation due to heterogeneity of PDE datasets and (b) too many parameters necessary for dense models to alleviate the previous challenge. The authors pre-train the DPOT with MOE on 6 PDE datasets and demonstrate zero-shot and fine-tuned performance on several downstream tasks. They show that the MOE addition uniformly helps increase the accuracy of DPOT. Further, they show that their MOE activations correspond to learning the underlying PDE task distribution, motivating its creation. Some ablations on number of experts demonstrate that a balance is being struck w.r.t accuracy vs speed. Finally, the authors show the practical utility of their model by comparing inference times - the setting where these models will be used in production.

**Questions:**

Clarity:
* The results are unclear in some places. It would be useful to demonstrate clearly what is the train/val/test split for each dataset used, which datasets are used for pre-training vs zero-shot vs finetuning, size of the datasets for each case (pre-training/finetuning), degree of difficulty for the downstream tasks w.r.t out-of-distribution generalization (this can be as simple as quantifying the effects of the physics coefficients on the difficulty of the tasks -  some tasks such as diffusion would be easier to generalize rather than more hyperbolic systems and hence the error metrics are not sufficient to gauge the improvements).
* Tab1. would be more beneficial as some sort of figure (error vs fine-tuning dataset size) - it's then clear what value the MOE adds in addition to the dataset volume.
* Fig 2 (right) would be beneficial to see it as one figure (with different color bars) so it's easier to compare the MOE activations

Experiments:
* The error over DPOT seems very comparable. How many finetuning examples are taken? How much care is given to optimization when finetuning to a new dataset (learning rates, batch size, frozen weights)? Will DPOT eventually catch-up to MOE with more examples and at what point (how many examples)? Finally, I understand that dropping the error from 0.02 to 0.01 is a 100% improvement, but how valuable is that if your error is already low? It seems like DPOT does get low errors already in most cases. I think it would be beneficial to stress-test on more out-of-distribution examples.
* I appreciate the interpretability experiment. I feel like the authors could do more with this line of analysis for this paper. For example, when you look at classification accuracy, is it just for a training sample X or validation/test? How does the accuracy change as you pick X from a downstream task and systematically make that downstream task out-of-distribution? Do the MOE logits still point to some pre-training dataset or does it become more uniform?
* Related to above, the authors mention that part of the MOE is to learn common representations (conservation laws) and the other part is to learn the PDE task distribution? Is there any evidence to suggest the former? Such as high activation for the shared experts for all datasets? Similarly, if you picked X (input) from an out-of-distribution example, would you still see the shared experts being activated? That would be a nice result.
* Another related thread: if part of the MOE is just to distinguish the PDE, why not just one-hot encode that information at the beginning (or have the MOE as well at the beginning). It's understandable in fields like language modeling that the source of the tokens is pretty unclear but the science, data is often from structured simulation+experiments and hence it should be possible to encode that information directly?
* The inference time experiment is not convincing. DPOT-M and MOE-M have roughly the same inference time. Tab 1 only compares the accuracy of these two models (as the best models), so I'm wondering what the advantage of MOE is? Also what is the accuracy of DPOT-L in Tab. 1?
* How does the MOE block impact rollout accuracy? This probably is more important than one-step prediction accuracy for spatiotemporal models. It would be good to discuss the multistep performance as well.
* [Minor] What is the performance of the FNO with the MOE? The FNO numbers seem disentangled from the whole experiment and seem to only emphasize that DPOT is a better architecture choice?

Minor:
* Fig labels and tick sizes could be made larger so it's clear
* Tab 1. The caption says error greater than 20. I did not follow what that means
* What is n in W_p (after Eqn (3))? embedding dimension?

**Ethical Concerns:**

["NO or VERY MINOR ethics concerns only"]

**Final Justification:**

The paper presents a novel contributed to PDE foundation models using MoE. The performance increase is marginal but is still an interesting analysis to the community. There are some clarity revisions that the authors have committed too along with new results from the discussion phase from all reviews. Overall, I recommend a borderline accept

**Limitations:**

There is a minimal discussion on limitations.
* It seems like a limitation that the MOE is evaluated on simple and small pre-training volumes (6 datasets from PDEBench). I would expect that evaluations on larger volumes such as The Well (https://polymathic-ai.org/the_well/datasets_overview/) would show up more interesting ways the MOE could benefit the training or show some instabilities with the large data distributions. Some discussion on this would be useful.

**Paper Formatting Concerns:**

-

**Quality:**

2

**Strengths And Weaknesses:**

Strengths:
* The paper is well-motivated, the topic is of high interest to the NeurIPS audience
* Performance of the MOE model is better than DPOT (a SOTA pre-training architecture for SciML)
* Experiments demonstrate that the experts learn the PDE data in-distribution structure by looking at the MoE activation distributions

Weaknesses:
* The paper's clarity could be improved (see comments). While the motivation is clear, the results section could benefit greatly with more clarifications
* The improvement from DPOT seems marginal and limited to one-step improvement. It's unclear how much this translates to rollout improvements
* There is very little discussion on out-of-distribution generalization properties of the MOE model (or others)
* The interpretability analysis (while appreciated) could be made more extensive to stress-test the motivation behind the MOE (see comments)

---

> ### Author Rebuttal · Authors · 2025-07-31
>
> Dear Reviewer,
>
> Thank you for your detailed and insightful review. We are very grateful for your positive feedback on our paper's motivation and the novelty of our approach. Your comments are exceptionally helpful, and we agree that they point to clear ways we can improve the paper. We have carefully considered all your questions and provide the following responses, including new experimental results, which we hope will address your concerns.
>
> ---
>
> ### **On Clarity and Experimental Details**
>
> We completely agree that presenting experimental details with utmost clarity is crucial. Thank you for these excellent suggestions. We have uploaded our code for full transparency.
>
> - **Experimental Setup & Data Splits:** As stated on Page 6 (Line 186), our datasets are from three widely-used public benchmarks: FNO, PDEBench, and CFDBench. We used the standard, publicly available train/validation/test splits and sample sizes for all benchmarks to ensure fair and reproducible comparisons, following the standard procedures of prior works like DPOT and FNO.
> - **Commitment to Improvement:** We recognize that these details could be presented more clearly. In the final version, we will:
>   - **Add a comprehensive summary table** in the appendix, detailing the train/val/test splits, sample counts, fine-tuning procedures, and data sources for all pre-training and downstream datasets. We will also include quantitative metrics of task difficulty (e.g., Reynolds number for NS equations) and discuss their implications for generalization.
>   - **Revise Figure 2 (right)** into a single, unified plot with grouped bar charts. This will allow for a direct and easy comparison of expert activation patterns across different datasets, as you wisely suggested.
>   - **Improve Table 1's presentation** by adding a summary figure in the main text that plots performance against the number of fine-tuning samples for MoE-POT vs. DPOT, visually highlighting the value added by our MoE architecture. The full table will be retained in the appendix.
>   - **Enlarger all figure labels and ticks** to improve readability.
>
> ### **On Performance Improvement and Multi-Step Rollout Accuracy**
>
> You have raised a crucial point about the significance of our performance gains and the fairness of our comparisons. We would like to offer a detailed clarification.
>
> - **Reported Errors are for Multi-Step Rollouts:** We would like to clarify a critical point regarding our evaluation: the reported errors in our tables are **not for single-step prediction but are the average errors over long-term, auto-regressive rollouts.** Our experimental setup follows the standard protocol used in DPOT and FNO. For instance, on the PDEBench-SWE dataset, we input the first 10 frames and auto-regressively predict the subsequent 90 frames, with the reported error being the average over these 90 predicted steps.
> - **The Value of "Marginal" Error Reduction:** For spatio-temporal dynamics, even a small reduction in single-step prediction error is vital.
>   - **Significant Relative Improvement:** As shown in Tables 1 and 2, our model achieves a substantial *relative* performance improvement on multiple tasks. In the field of neural operators, where current SOTA precision (1e-2 to 1e-4) still lags behind traditional solvers (<1e-7), every bit of error reduction is a meaningful step forward.
>   - **Importance of Compounded Error:** A significant reduction in single-step error (e.g., 50% relative improvement) drastically suppresses the accumulation of errors during long-term rollout predictions. This leads to far more stable and accurate simulations in practical applications, directly addressing your concern about multi-step performance.
> - **Fair Comparison with DPOT:**
>   - **Fine-tuning Details:** As described on Page 7 (Lines 221-222) and Page 8 (Line 240), our fine-tuning protocol was applied consistently across all pre-trained models (DPOT and ours) to ensure a fair comparison. For Table 1, models were fine-tuned for 200 epochs; for Table 2, 500 epochs. The training set of the corresponding dataset was used for fine-tuning.
>   - **Convergence** **and Architectural Advantage:** In all fine-tuning experiments, both MoE-POT and DPOT models reached convergence (typically before 180 epochs for Table 1 tasks and before 400 epochs for Table 2 tasks). Therefore, there was no possibility of DPOT "catching up" with more training. We argue the performance gap stems from a fundamental architectural advantage. As a dense model, DPOT's capacity is limited by its activated parameters. MoE-POT, through sparse activation, leverages a much larger total parameter count, giving it a higher capacity to absorb heterogeneous knowledge more effectively.
>
> ### **On Inference Efficiency and DPOT-L Comparison**
>
> - **Advantage in Inference Efficiency:** You correctly observe that MoE-POT-M and DPOT-M have similar inference times. This is precisely where our model's advantage lies. As shown in Table 3, MoE-POT-M (288M activated, 489M total params) has nearly the same inference time (16.6ms) as DPOT-M (158M activated, 16.7ms). However, as shown in Table 1, MoE-POT-M is far more accurate for this same inference cost. This demonstrates a significantly better efficiency-to-performance ratio and highlights the scalability advantage of the MoE architecture.
> - **DPOT-L Experimental Results:** Due to computational constraints, we were unable to complete the full pre-training for a corresponding MoE-POT-L model. To address your question, we provide the zero-shot results for the official DPOT-L model below.
>
> | Model and Activation Parameter | NS(1e-5) | NS(1e-3) | CNS(0.1,0.01) | SWE    | DR     | CFDBench |
> | --- | --- | --- | --- | --- | --- | --- |
> | DPOT-S 31M      | 0.0078   | 0.0688   | 0.0244        | 0.0039 | 0.0367 | 0.0087   |
> | DPOT-M 122M      | 0.0071   | 0.0569   | 0.0224        | 0.0025 | 0.0288 | 0.0113   |
> | DPOT-L 493M       | 0.0061   | 0.0576   | 0.0113        | 0.0023 | 0.0219 | 0.0065   |
> | MoE-POT-T 17M    | 0.0077   | 0.0682   | 0.0105        | 0.0064 | 0.0411 | 0.0053   |
> | MoE-POT-S 90M      | 0.0058   | 0.0552   | 0.0096        | 0.0029 | 0.0342 | 0.0045   |
> | MoE-POT-M 288M      | 0.0057   | 0.0528   | 0.0091        | 0.003  | 0.03   | 0.0051   |
>
> These results show that **MoE-POT-M achieves comparable or superior performance to DPOT-L on 4 out of 6 datasets, using only about half the activated parameters (and thus, computational cost).** This strongly supports our claims. We will add a more detailed comparison with DPOT-L in the final version.
>
> ### **On Interpretability and the MoE Mechanism**
>
> Thank you for appreciating the interpretability analysis and for pushing us to explore it further. These are excellent suggestions.
>
> - **Role of Shared Experts:** The two shared experts are, by design, activated for every input. In our initial experiments, models without shared experts often ended up training unstable.
> - **Stress-Testing with Out-of-Distribution Data:** In our original analysis (Appendix B.3), we used the training set to compute the mean expert preference vector for each dataset and the test set to calculate classification accuracy. To address your excellent question about OOD data, we conducted a **new experiment** testing the classification accuracy on two challenging downstream tasks from the Poseidon paper [1] that were **not seen during pre-training**.
>
> | Dataset (Unseen) | Block-1 Accuracy | Block-2 Accuracy |
> | ---------------- | ---------------- | ---------------- |
> | Wave-Layer       | 100%             | 100%             |
> | Wave-Gauss       | 100%             | 100%             |
>
> The router achieves **100% classification accuracy** on these unseen datasets. Furthermore, in Block-2, the expert network ratios for the Wave-Layer dataset were: Expert 1 (79%), 9 (31%), 11 (100%), 13 (100%), 15 (100%), others (0%). This strongly demonstrates that the gating network learns to distinguish fundamental properties of different PDEs and generalizes this capability to new, unseen tasks.
>
> ### **Why Not One-Hot Encode?:**
>
> - Our work follows the experimental paradigm and dataset of DPOT and FNO, which aims to simulate real-world scenarios (e.g., weather forecasting) where only solution trajectories are available, and the exact PDE parameters are unknown. The goal is for the model to implicitly infer the system's dynamics from observed data, which is precisely the role our router network successfully plays. While parameter-informed modeling via encoding is a valid and important research direction, our core contribution is the effective introduction of the MoE structure into this auto-regressive pre-training paradigm.
>
> ### **On Minor Questions**
>
> - **FNO with MoE:** During our research, we explored numerous combinations, including MoE with FNO and FFNO. These attempts were unsuccessful, leading to unstable training and large errors that could not be resolved by fine-tuning. The current MoE-POT architecture was the final stable and effective design that emerged from this rigorous process.
> - **Table 1 Caption:** Thank you for pointing this out. The "-" indicates that the model's L2RE on that task exceeded 20, signifying that training diverged or failed completely. We will clarify this in the caption.
> - **Wp in Eqn. (3):** You are correct. 'n' is the feature dimension of the positional encoding, which we typically set to be the same as the number of channels C. We will explicitly define this in the final version.
>
> **References:**
>
> [1] Poseidon: Efficient foundation models for pdes, NeurIPS 2024.
>
> ---
>
> #### **Thanks again**
>
> We sincerely thank you again for your constructive and detailed feedback, which has helped us identify clear pathways to improve our paper. Should you have any further questions or require additional discussion, please don't hesitate to reach out. If we have adequately addressed your concerns, we would be grateful for your consideration in adjusting your evaluation score accordingly.

---

> > ### Comment · Reviewer_fgJS · 2025-08-05
> >
> > Thank you authors for the detailed responses, they have mostly addressed my questions. Also, thank you for the commitment to improve the presentation of the paper, I think this is important for any revision.
> >
> > I appreciate the OOD stress-tests and the new experiments from the other reviews.
> >
> > Follow-up clarifications:
> > 1. Rollout errors: I am confused by the phrasing of the response. Is Tab 1. the rollout error on multiple steps or is your point that it is a single step rollout and small reductions would imply that multistep rollout error would also decrease? I am going to assume it is the former - please do emphasize in the table captions and text since I'm not sure a reader is always going to be familiar with the evaluation protocols of benchmark datasets.
> > 2. Finetuning accuracy comparison: the response reads "In all fine-tuning experiments, both MoE-POT and DPOT models reached convergence (typically before 180 epochs for Table 1 tasks and before 400 epochs for Table 2 tasks). Therefore, there was no possibility of DPOT "catching up" with more training."
> > I think my question was if DPOT would catch up with more finetuning examples. It still seems like the MoE integration leads to marginal (but as the authors correctly point out, useful) gains - hence, I wonder if a few more data points would close this gap (and not training for more epochs). This would be a marginal cost offloaded to fine-tuning.
> > The response seems to focus on number of epochs used for finetuning whereas my question is number of examples used. Could you please clarify?

---

> ### Author Response · Authors · 2025-08-06
> **Follow-up Response: New Experimental Results and Clarifications**
>
> Dear Reviewer,
>
> Thank you for your prompt follow-up and for giving us the opportunity to provide further clarification. We apologize that our previous response was not entirely clear due to length constraints. We are grateful for the chance to address your remaining questions in more detail.
>
> ---
>
> ### **On the Nature of Rollout Errors (Q1)**
>
> To clarify directly: **all results reported in our paper, including those in Table 1, are indeed multi-step rollout errors.** We apologize if our previous phrasing caused any confusion and would like to explain our evaluation protocol precisely.
>
> Our methodology, consistent with the DPOT benchmark, involves an auto-regressive rollout. For example, on the PDEBench-SWE dataset, we input the solution function images for frames 1, 2, ..., 10. We have the model predict frame 11. We then use frames 2, 3, ..., 10, 11 (the newly predicted frames) to predict frame 12. We then use frames 3, 4, ..., 10, 11 (the newly predicted frames), 12 (the newly predicted frames) to predict frame 13. This cycle continues until we reach frame 100. We then use the average L2RE error over frames 11, 12, ..., 100 as the model's error on the current dataset. This evaluation method is used in all experiments in the paper (including all tables), and is also the evaluation method used in the DPOT experiments.
>
> We believe this process aligns with what you refer to as "multi-step rollout", where each prediction in the sequence is a "single-step prediction".You are absolutely right that a small reduction in single-step error leads to a significant reduction in the cumulative multi-step error. This effect is very pronounced because errors compound with each new prediction. To illustrate this with the SWE dataset:
>
> | Model     | Frame 50 L2RE | Frame 70 L2RE | Frame 100 L2RE | Average L2RE |
> | --- | --- | --- | --- | --- |
> | DPOT-S    | 0.0031        | 0.0034     | 0.0051         | 0.0039       |
> | MoE-POT-S | 0.0024        | 0.0026     | 0.0035         | 0.0029       |
>
> As the table shows:
>
> 1. The error for both models increases with more rollout steps, confirming the effect of error accumulation.
> 2. The performance gap between DPOT-S and MoE-POT-S widens from a ~25% relative difference at frame 50 to a ~45% difference at frame 100. This super-linear divergence in error underscores the critical impact of reducing single-step error.
>
> Your suggestion is invaluable for improving the paper's clarity. **We commit to explicitly stating this evaluation protocol in the main text and table captions in the final version. We will also add this detailed breakdown of per-frame error analysis to the appendix to avoid any ambiguity for future readers.**
>
> ---
>
> ### **On Fine-tuning Performance vs. Number of Examples (Q2)**
>
> We sincerely apologize for misinterpreting your previous question. **Your intuition is spot on: increasing the number of fine-tuning samples can indeed allow a model like DPOT to eventually match the performance of MoE-POT.**
>
> To address your question directly, we conducted new experiments to illustrate this relationship. We tested performance on two tasks: a downstream task closely related to our pre-training data (NS 1e-4) and an out-of-distribution task from Poseidon [1] (Wave-Layer).
>
> - Experiment 1: In-Distribution Fine-tuning on NS (1e-4) (500 Epochs)
>
> | Number of samples | 16   | 32   | 64   | 128  | 512   | 2000  |
> | ---- | ---- | ---- | ---- | --- | ---- | ---- |
> | DPOT-S        | 0.25 | 0.20 | 0.13 | 0.09 | 0.044 | 0.026 |
> | MoE-POT-S         | 0.20 | 0.15 | 0.11 | 0.07 | 0.040 | 0.016 |
>
> - Experiment 2: Out-of-Distribution Fine-tuning on Wave-Layer (500 Epochs)
>
> | Number of samples | 16   | 32   | 64   | 128  |
> | --- | ---- | --- | --- | --- |
> | DPOT-S            | 0.41 | 0.33 | 0.26 | 0.19 |
> | MoE-POT-S         | 0.34 | 0.26 | 0.20 | 0.14 |
>
> As demonstrated in the experiments above, and as you might expect, model performance improves on downstream tasks as the number of samples increases. Therefore, DPOT with more samples will outperform MoE-POT. In this scenario, a high-performing model requires less data to achieve similar performance, or achieves better performance with the same number of samples. Therefore, in all downstream task experiments in this article, we use a fixed number of samples.
>
> **We are very grateful for this insightful question and will incorporate these new results and analysis into the final version of our paper.**
>
> References:
>
> [1] Pdebench: An extensive benchmark for scientific machine learning, NIPS 2022.
>
> ---
>
> We sincerely thank you once again for your detailed and constructive engagement with our work. We hope these detailed explanations fully address your remaining concerns. Should you have any further questions, please do not hesitate to let us know. We remain available for any further discussion. We are very grateful for your time and guidance, and we would be deeply appreciative if you would consider our clarifications in your final evaluation of our work.

---

> > ### Comment · Reviewer_fgJS · 2025-08-06
> >
> > Thank you very much authors, the clarifications are much appreciated.
> >
> > Finetuning: I guess this answers my questions - it seems like you will pay a 2x data cost at finetuning to match the performance of your MoE model. I think this can be said in the paper because it captures an important trade-off.
> > It is interesting that the OOD data costs are higher (and what is probably a nice result), whereas in in-distrubution, DPOT sort of catches up. Maybe this is also nice to show (you could use figures with log scales with accuracy vs number of finetuning samples with zero-shot at a constant line to it's clear to the reader what the tradeoff is).
> >
> > At this point, I echo reviewer frLV 's comments on revision clarity and the marginal performance increase as the negative side, but overall, raise my score to 4 and hand it over to the AC for the decision.

---

> > > ### Author Response · Authors · 2025-08-07
> > > **Thanks for Your Feedback and Revision Plan**
> > >
> > > We greatly appreciate your constructive feedback and your willingness to raise your score!
> > >
> > > As suggested, we will revise the paper to explicitly discuss the finetuning data cost trade-off. We will also add the new figure you proposed to clearly visualize this comparison.
> > >
> > > Thank you again for your valuable suggestions to improve our paper.

---

### Official Review · Reviewer_wk8j · 2025-07-02

**Clarity:** 3
**Significance:** 3
**Originality:** 3
**Rating:** 5
**Confidence:** 5

**Summary:**

This paper trains a mixture of experts model for surrogate PDE models. Since different PDEs have different types of dynamics, initial conditions as well as boundary conditions, the authors show that training a mixture of experts model where a router routes different pdes to different experts (along with having two shared experts). The authors show that their model with 90M active parameters improves upon the zero-shot performance of baselines, and show that different experts do end up taking in PDEs of similar form, and therefore the type of the PDE data can potentially be inferred from the expert that the model ends up going to.

The architecture in this paper is similar to the DPOT architecture with Fourier attention layers with appropriate temporal aggretation layers. For the mixture of experts and to maintain sparsity, the Top-K activations are used. Appropriate regularization is applied to ensured that not all the examples go through the same tokens (Load balancing objective).

The authors train their method on 5 datasets from PDEBench and CFDBench., and show that their model consistently outperforms multiphysics baselines like DPOT, as well as single model per dataset type of baselines like FNO.

**Questions:**

the authors mention something to the form of “model leads to feature competition” and “knowledge illusion” in lines 95-96. What do they mean by that, and more importantly, can they cite relevant works that coin these terms?

**Ethical Concerns:**

["NO or VERY MINOR ethics concerns only"]

**Final Justification:**

The authors have address all my concerns and also added extra experiments, and have comparisons with more recent multi-physics models. I thank the authors for the effort and have also updated my scores accordingly.

**Limitations:**

Yes

**Quality:**

3

**Strengths And Weaknesses:**

The paper is easy to read and follow.

I think that this is perhaps the first MOE papers for PDE solvers, something that makes a lot of sense, since as the authors mention, each PDE has different dynamics, and variability through initial and boundary conditions.

The fact that similar PDEs are routed through same experts, and that this emerges is quite promising, and interesting.

However, there are few points that make me question a lot of the contributions of the paper.

- First of all, the multi-physics models that they try are DPOT, and not models like Poseidon and MPP (especially Poseidon since they are currently the best performing multiphysics models).
- The authors compare their numbers with FNO, but its a very old model and since then a lot of improvements have been suggested on top of FNO (such as FFNO, and others) that could have better performance over FNO.
- The datasets that the authors train on, like PDE bench are not very difficult, and it is quite easy to get a good models on these datasets, therefore it is unclear how good the architecture is here.
- The authors mention that they finetune the model for 200 epochs, that is quite a lot, esp since the model has been pre-trained. The authors do show that training from scratch does not work as much (in Table 2), but for single model per dataset baselines like FNO (and its better performing derivatives like FFNO), it would be interesting to see how well it performs if it is given the overall compute budget (of pretraining + finetuning).

---

> ### Author Rebuttal · Authors · 2025-07-31
>
> Dear Reviewer,
>
> Thank you for your thorough review and positive assessment of our work. We are grateful for your insightful feedback and for recognizing the novelty and promise of applying a Mixture-of-Experts (MoE) architecture to PDE solvers. Your questions have been instrumental in helping us clarify our contributions. We have carefully considered your points and provide the following detailed responses.
>
> ---
>
> ### **On the Choice of Baselines (DPOT vs. Poseidon/****MPP****) (Weakness 1)**
>
> Thank you for this valuable suggestion. We chose DPOT as our primary multi-physics baseline for three main reasons:
>
> - **Clear Attribution of Gains:** Our MoE-POT architecture is built upon the DPOT model, where we specifically replace the dense feed-forward network with our sparse expert layer. This design allows us to cleanly attribute performance differences to the advantages conferred by the MoE architecture itself, providing a direct and compelling validation of our core method. Comparing with structurally disparate models like Poseidon would make it difficult to isolate the source of any performance gains.
> - **Differences in Experimental Setup:** Our work, following DPOT, uses an auto-regressive paradigm: predicting the next frame from a sequence of previous frames. In contrast, models like MPP and Poseidon have different setups. For instance, Poseidon takes problem parameters, boundary conditions, and initial conditions as input to predict a future state. As noted in the original DPOT paper, DPOT outperformed MPP in most scenarios. Furthermore, Poseidon requires PDE parameters as input, which are not available in the public datasets we use (FNO, PDEBench, CFDBench), making a direct comparison infeasible under our experimental protocol.
> - **Contemporaneous Work:** Poseidon is an excellent and cutting-edge model. At the time we began our experimental cycle, this work had not yet been publicly released and was developed concurrently with our own. Therefore, we were unable to include it in our initial baseline comparisons.
>
> We agree that comparison with the latest state-of-the-art models is crucial. To address your concern, we conducted supplementary fine-tuning experiments on two challenging downstream tasks from the Poseidon paper, Wave-Layer and Wave-Gauss, using the $L_2$ relative error metric. We considered two settings:
>
> 1. **Our Setting (Auto-regressive, no parameters):**
>
> | Model (Activated Params) | Wave-Layer | Wave-Gauss |
> | ------------------------ | ---------- | ---------- |
> | Poseidon-T (21M)         | 0.29       | 0.29       |
> | Poseidon-B (158M)        | 0.21       | 0.24       |
> | MoE-POT-T (17M)          | 0.07       | 0.07       |
> | MoE-POT-S (90M)          | 0.05       | 0.06       |
>
> 2. **Poseidon's Setting (Parameter-informed):**
>
> | Model (Activated Params) | Wave-Layer | Wave-Gauss |
> | ------------------------ | ---------- | ---------- |
> | Poseidon-T (21M)         | 0.08       | 0.06       |
> | Poseidon-B (158M)        | 0.06       | 0.09       |
> | MoE-POT-T (17M)          | 0.11       | 0.14       |
> | MoE-POT-S (90M)          | 0.06       | 0.10        |
>
> These results indicate that each model excels in its native experimental setting, suggesting they have different scopes of application rather than one being definitively superior. Due to the time constraints of the rebuttal period, we could not complete an exhaustive comparison. However, if the paper is accepted, **we commit to including a more thorough comparison with Poseidon and other recent** **SOTA** **models in the final version**, along with citations to these important works.
>
> ### **On Comparison with Advanced FNO Variants (Weakness 2)**
>
> Thank you for your close reading. We completely agree that a comparison limited to the original FNO would be insufficient. In fact, in **Table 1 of our paper, we have already conducted a comprehensive comparison against a range of more advanced single-task baselines.** This includes not only the **FFNO** you mentioned but also **GK-Transformer, OFormer, and GNOT**. The results demonstrate that our fine-tuned MoE-POT model outperforms these state-of-the-art single-task baselines on the vast majority of datasets, validating the effectiveness of our pre-training framework.
>
> ### **On the Difficulty of Pre-training Datasets (Weakness 3)**
>
> We appreciate your concern regarding the choice of datasets. We selected them based on the following considerations:
>
> - **Standard and Fair Benchmarking:** The six datasets we use are sourced from three widely-accepted public benchmarks: FNO, PDEBench, and CFDBench. They are also the datasets used for pre-training by DPOT. Using these standard benchmarks ensures that our results are fair, reproducible, and comparable with existing and future research.
> - **The Core Challenge is "****Heterogeneity****," Not "Single-Task Difficulty":** The central thesis of our paper is to solve the performance bottleneck that arises when a single model must handle a *heterogeneous mixture* of PDEs. As we explicitly demonstrate in **Figure 2 (left)**, even on these individually "less difficult" datasets, mixing them during training causes the performance of a traditional dense model to collapse. Therefore, the advantage of our architecture is its ability to effectively manage data diversity and heterogeneity, a property that is best demonstrated through the diverse dataset combination we selected.
>
> ### **On Fine-Tuning Epochs and the Overall Compute Budget (Weakness 4)**
>
> This is a very insightful and fair point. We offer two clarifications:
>
> - **Relative Cost of Fine-tuning:** Our pre-training phase runs for 1000 epochs, making the 200-epoch fine-tuning a reasonable fraction of the total training. More importantly, the value of fine-tuning lies in its ability to rapidly adapt general knowledge to specific tasks. The results in **Table 2** clearly show that pre-trained models ("w/ Pre-train") significantly outperform models trained from scratch ("w/o Pre-train") on all downstream tasks, proving the immense value and knowledge transfer capability of pre-training.
> - **Model** **Capacity** **vs. Training Duration:** The idea of aligning the total compute budget is an excellent one. However, we argue that smaller models like FFNO, even if granted equivalent training time, would quickly hit a performance ceiling due to their limited model capacity, potentially even degrading from overfitting. Indeed, all baseline results in our paper represent the best performance achieved during their training until full convergence. In contrast, our MoE architecture's success stems from having a very large total parameter count (ensuring high capacity) while keeping computational costs low via sparse activation. Our scaling experiments (Sec 5.3) further support the conclusion that larger model capacity leads to better performance.
>
> ### **Clarification of "Feature Competition" and "Knowledge Illusion" (Question)**
>
> Thank you for pointing out this imprecise phrasing. These terms were intended as intuitive descriptors for the well-established phenomenon known as **"****Negative Transfer****"** in multitask learning.
>
> - **"Feature Competition"** refers to the scenario where a dense model, forced to learn fundamentally different tasks (e.g., fluid dynamics of NS equations vs. reaction-diffusion of DR equations) with a single set of shared parameters, experiences conflicting gradient updates. This "tug-of-war" between tasks prevents the model from fully learning any single task, thereby harming overall performance.
> - **"Knowledge Illusion"** describes the outcome of this phenomenon: the model appears to have learned something about all tasks, but this knowledge is shallow and compromised, failing to capture the core physics of each task and leading to poor generalization.
>
> We acknowledge that we did not provide direct citations for these descriptive terms. In the final version of the paper, we will replace them with the more standard academic terms **"****negative transfer****"** or **"inter-task interference"** and cite foundational multitask learning literature, such as [1], to make our argument more rigorous.
>
> **References:**
>
> [1] Multitask learning, Caruana, 1997
>
> ---
>
> #### **Thanks again**
>
> We sincerely thank you again for your constructive and detailed feedback, which has helped us identify clear pathways to improve our paper. Should you have any further questions or require additional discussion, please don't hesitate to reach out. If we have adequately addressed your concerns, we would be grateful for your consideration in adjusting your evaluation score accordingly.

---

> ### Author Response · Authors · 2025-08-03
> **Detailed Supplement on Baseline Comparisons (Weakness 1)**
>
> We agree that a robust comparison with state-of-the-art models like Poseidon is crucial for positioning our work. Due to the time constraints of the rebuttal period, our initial experiments were incomplete. We have now finalized our supplementary experiments comparing MoE-POT with Poseidon, and we are happy to report that our initial conclusions remain consistent.
>
> ---
>
> We conducted comparisons under two distinct settings to ensure a comprehensive evaluation:
>
> **1. Poseidon's Setting**
>
> To align with the methodology in the Poseidon paper, we modified MoE-POT’s architecture to match their input-output structure (i.e., taking problem parameters and initial conditions as input to predict a specific future frame). For this comparison, we fine-tuned each model on 2,000 trajectories for 200 epochs.
>
> | Model (Activated Params) | Training Trajectories | Wave-Layer (L2RE) | Wave-Gauss (L2RE) |
> | ------------------------ | --------------------- | ----------------- | ----------------- |
> | Poseidon-T (21M)         | 2000                  | 0.08              | 0.06              |
> | Poseidon-B (158M)        | 2000                  | 0.06              | 0.09              |
> | MoE-POT-T (17M)          | 2000                  | 0.11              | 0.14              |
> | MoE-POT-S (90M)          | 2000                  | 0.06              | 0.10               |
>
> **2. Our Setting**
>
> For this setting, we used the official open-source Poseidon repository and adapted its loss calculation to fit our auto-regressive framework. All models were trained for 200 epochs.
>
> In our initial analysis, we overlooked that the official Poseidon training script defaults to using only 128 trajectories, which resulted in an unfair comparison. We sincerely apologize for this oversight. We have now included a more complete and equitable comparison, detailing the performance of MoE-POT-S with varying numbers of training trajectories to demonstrate its effectiveness in low-data regimes.
>
> | Model (Activated Params) | Training Trajectories | Wave-Layer (L2RE) | Wave-Gauss (L2RE) |
> | ------------------------ | --------------------- | ----------------- | ----------------- |
> | Poseidon-T (21M)         | 128                   | 0.29              | 0.29              |
> | Poseidon-B (158M)        | 128                   | 0.21              | 0.24              |
> | MoE-POT-T (17M)          | 2000                  | 0.07              | 0.07              |
> | MoE-POT-S (90M)          | 16                    | 0.34              | 0.37              |
> | MoE-POT-S (90M)          | 32                    | 0.26              | 0.28              |
> | MoE-POT-S (90M)          | 64                    | 0.20             | 0.20              |
> | MoE-POT-S (90M)          | 128                   | 0.14              | 0.13              |
> | MoE-POT-S (90M)          | 2000                  | 0.05              | 0.06              |
>
> These updated and expanded results compellingly show that each model excels within its native experimental setting. This suggests they are powerful tools with different, valuable scopes of application rather than one being definitively superior across all paradigms. Due to the time constraints of the rebuttal period, we could not complete an exhaustive comparison. However, if the paper is accepted, we commit to including a more thorough comparison with Poseidon and other recent SOTA models in the final version, along with citations to these important works.

---

> ### Author Response · Authors · 2025-08-06
> **To Reviewer wk8j: Thanks and Follow-up Discussion**
>
> Dear Reviewer wk8j,
>
> Thank you very much for taking the time to read our rebuttal and for promptly acknowledging it. We truly appreciate your engagement with our work during the review process.
>
> If you have any further thoughts, questions, or suggestions---either regarding our current response or the broader direction of the work---we would be genuinely grateful to hear them. We highly value your perspective and would welcome any opportunity for continued discussion or clarification.
>
> Best,
>
> Authors

---

### Official Review · Reviewer_J1rL · 2025-07-03

**Clarity:** 3
**Significance:** 2
**Originality:** 2
**Rating:** 4
**Confidence:** 4

**Summary:**

This paper introduces MoE-POT, a Mixture-of-Experts Pre-training Operator Transformer, which modifies the MLP Layer of a DPOT[1] model using a router‐gating network that dynamically selects 4 routed experts from 16 total, alongside 2 shared experts, per layer. Pre-trained on six public PDE datasets with total parameters ranging from 30 M to 0.5 B (activated parameters 17 M–288 M), MoE-POT achieves up to 40% reduction in zero-shot L₂ relative error compared to existing dense DPOT models and infers input PDE types with 98% accuracy via gating patterns.

[1] "DPOT: Auto-regressive denoising operator transformer for large-scale pde pre-training" Hao et al, ICML 2024

**Questions:**

* The paper claims that MoE-POT-M(same number of parameters as DPOT-L) which has a larger number of activated parameters than DPOT-M are equivalent. While inference time reported is similar, the accuracy improvement over DPOT-M seems to come at a significant memory and training cost. Moreover a comparison with DPOT-L is not provided. Can the authors provide a detailed analysis of the training and memory overhead and also add the DPOT-L results?
* Does MoE effectively handle diverse PDE datasets as claimed by the paper? An ablation study showing performance gains with increasing number of datasets as compared to a normal DPOT would justify this claim.
* It seems like the MLP layer of other transformer-based PDE foundation models like MPP[2] can also be replaced by an MoE layer resulting in a similar architecture. What makes MoE-POT unique?

[2] "Multiple Physics Pretraining for Physical Surrogate Models" McCabe et al. NeurIPS 2024

**Ethical Concerns:**

["NO or VERY MINOR ethics concerns only"]

**Final Justification:**

The rebuttal answers a lot of my questions. I have been following the other discussions as well and MoE-POT does seem to provide a small edge over DPOT. Thus I raise my score to acceptance

**Limitations:**

Yes

**Quality:**

3

**Strengths And Weaknesses:**

### Strengths:
* Interpretability: The router-gating network reliably encodes dataset identity showing that expert selections reflect underlying PDE types
### Weakness:
* Novelty: The architecture is just DPOT with the MLP layer replaced with an MoE, which is not enough novelty in the architecture.
* I don't see a clear win for using an MoE layer in the architecture. The accuracy gain seems to come at a huge cost in terms of memory and training cost.
* Lack of Theoretical Analysis: The paper claims that router-gating decisions yield dataset classification but does not analyze the mathematical underpinnings or guarantees of this mechanism which is noted in limitations.

---

> ### Author Rebuttal · Authors · 2025-07-31
>
> Dear Reviewer,
>
> Thank you for your thoughtful and constructive feedback. We appreciate your time and have carefully considered your suggestions. We hope the following clarifications address your concerns and further highlight the significance of our work.
>
> ---
>
> ### **On the Novelty and Uniqueness of MoE-POT (Weakness 1 & Questions 3)**
>
> Our core innovation is not merely substituting a layer but systematically addressing the critical bottleneck of data heterogeneity in large-scale PDE pre-training. MoE-POT is presented as a well-validated and highly interpretable solution to this specific, previously underexplored challenge.
>
> - **Problem-Driven Innovation:** Prior works have overlooked the "knowledge conflict" that arises from mixing diverse PDEs. Our experiments show this is a critical issue (Figure 2, left), as dense models suffer a drastic performance drop when trained on heterogeneous datasets.
> - **Unique and Validated Architectural Solution:** The uniqueness of MoE-POT stems from its "divide-and-conquer" strategy, enabled by the combination of routed and shared experts. The routed experts specialize in the unique characteristics of specific PDEs, while the shared experts capture common physical principles. Crucially, we validate this mechanism's effectiveness through interpretability analysis (Figure 2, right; Figure 4c), demonstrating that the model learns to dynamically select experts based on the PDE type with an accuracy of up to 98%.
> - **Rigorous Experimental Exploration:** The current MoE-POT architecture is the result of extensive experimentation. We explored numerous combinations of MoE with other neural operators (including MPP, FNO, and FFNO) and various MoE structures (e.g., without shared experts, using MLP-based experts instead of CNNs). These alternative designs frequently failed, resulting in models with substantial errors on certain datasets that could not be rectified through fine-tuning. The final version of MoE-POT is the stable and effective architecture that emerged from this rigorous process.
>
> Therefore, the novelty of MoE-POT is not in an isolated architectural tweak but in its systematic demonstration that a sparse expert architecture is an effective and interpretable solution to the data heterogeneity challenge in PDE pre-training, supported by comprehensive evidence that prior models have not explored.
>
> ### **On the Cost-Benefit Analysis and Comparison with DPOT-L (Weakness 2  & Question 1)**
>
> Thank you for your detailed questions regarding the model's efficiency and training costs. We would like to clarify the cost-benefit trade-off of the MoE architecture and address the comparison with DPOT models.
>
> - **Clarifying the Comparison:** You rightly point out the comparison between our MoE-POT-M (489M total, 288M activated) and DPOT-L (493M total & activated). Our key argument is that the MoE architecture decouples total parameters (storage cost) from activated parameters (computational cost), achieving superior efficiency.
> - **Greater Parameter Efficiency & Faster Inference**: Our MoE-POT-S (90M activated) outperforms the larger DPOT-M (122M activated) on several tasks. Furthermore, MoE-POT-M (489M total) has an inference time comparable to the much smaller DPOT-M and is 32% faster than the similarly sized DPOT-L, a critical advantage for iterative PDE solving.
> - **Training Overhead and DPOT-L Results:** We acknowledge that the large size of the L-series models made it challenging to complete the full pre-training for MoE-POT-L within our available computational resources, which is why a direct comparison was not included in the main table. We also recognize that MoE requires more memory to store the total parameters. However, we argue this is a highly valuable trade-off for achieving superior model performance, stronger generalization, and critically, higher inference efficiency. This rationale is widely accepted in other fields, such as large language models. In practical, large-scale deployments, memory usage can be optimized by loading shared expert networks that are used across multiple parallel models.
>
> To further address your concerns, we provide supplementary experimental results (L2RE) for DPOT-L below:
>
> | Model and Activation Parameter | NS(1e-5) | NS(1e-3) | CNS(0.1,0.01) | SWE    | DR     | CFDBench |
> | ------------------------------ | -------- | -------- | ------------- | ------ | ------ | -------- |
> | DPOT-S 31M                     | 0.0078   | 0.0688   | 0.0244        | 0.0039 | 0.0367 | 0.0087   |
> | DPOT-M 122M                    | 0.0071   | 0.0569   | 0.0224        | 0.0025 | 0.0288 | 0.0113   |
> | DPOT-L 493M                    | 0.0061   | 0.0576   | 0.0113        | 0.0023 | 0.0219 | 0.0065   |
> | MoE-POT-T 17M                  | 0.0077   | 0.0682   | 0.0105        | 0.0064 | 0.0411 | 0.0053   |
> | MoE-POT-S 90M                  | 0.0058   | 0.0552   | 0.0096        | 0.0029 | 0.0342 | 0.0045   |
> | MoE-POT-M 288M                 | 0.0057   | 0.0528   | 0.0091        | 0.003  | 0.03   | 0.0051   |
>
> As the table shows: 1) DPOT's performance improves marginally when scaling from 122M to 493M parameters. 2) When comparing DPOT-L (493M activated) with MoE-POT-M (288M activated), DPOT-L performs best on two datasets, while MoE-POT is superior on the other four. This indicates that MoE-POT achieves comparable, and in many cases better, performance with nearly half the activated parameters, which aligns with our claims. We will clarify this cost-benefit analysis and incorporate more detailed results for L-sized models in the final version of the paper.
>
> ### **On the Lack of Theoretical Analysis (Weakness 3)**
>
> We fully agree with your assessment and have candidly noted this in Section 6, "Limitations and Conclusions". The current work focuses on empirically demonstrating the feasibility and effectiveness of the MoE architecture for PDE pre-training. Our results, particularly the 98% dataset classification accuracy, provide strong empirical evidence for the mechanism that the router network can identify PDE types. Investigating the underlying mathematical principles and theoretical guarantees is a valuable and important direction for future research, which we are excited to pursue.
>
> ### **On Effectively Handling Diverse PDE Datasets (Question 2)**
>
> Thank you for this insightful question. We believe the existing evidence in our paper robustly demonstrates that MoE-POT handles heterogeneous datasets more effectively than dense models like DPOT, due to its inherent architectural advantages.
>
> - **Fundamental Limitation of Dense Models:**  Our work highlights a core weakness of dense models. As shown in Figure 2 (left), when training on PDEs with disparate properties, the performance of dense model FNO degrades sharply, with errors increasing by up to 5000%.
> - **MoE-POT's Architectural Advantage**: MoE-POT is explicitly designed to resolve this "knowledge conflict," and its router network intelligently identifies different PDE tasks and assigns them to specialized expert networks. Our interpretability analysis provides direct evidence for this mechanism: the router's selection weights alone can infer the input data's PDE type with 97.7% accuracy. This confirms that MoE-POT can effectively distinguish and process heterogeneous data in a way that dense architectures like DPOT cannot.
> - **Superior Performance as a Result:** This architectural superiority translates directly into leading performance. In the zero-shot comparison in Table 1, our MoE-POT-S (90M activated params) reduces the error on the PDEBench-CNS dataset by 57% compared to the larger DPOT-M (122M activated params) after pre-training on six heterogeneous datasets. This clearly shows MoE-POT learns more effectively from diverse data with lower computational cost.
>
> To further address your concern and illustrate the relationship between the number of datasets and performance for a dense model, we conducted the following supplementary experiment with DPOT-S, measuring its performance on the six core datasets as we increased the total number of pre-training datasets.
>
> |                        | NS(1e-5) | NS(1e-3) | CNS(0.1,0.01) | SWE    | DR     | CFDBench |
> | ---------------------- | -------- | -------- | ------------- | ------ | ------ | -------- |
> | DPOT-S (6 datasets)    | 0.0078   | 0.0688   | 0.0244        | 0.0039 | 0.0367 | 0.0087   |
> | DPOT-S (10 datasets)   | 0.0069   | 0.0663   | 0.0224        | 0.0037 | 0.0575 | 0.0146   |
> | DPOT-S (12 datasets)   | 0.0079   | 0.0739   | 0.0129        | 0.0105 | 0.0724 | 0.0075   |
> | MoE-POT-S (6 datasets) | 0.0058   | 0.0552   | 0.0096        | 0.0029 | 0.0342 | 0.0045   |
>
> These results show that: 1) Regardless of the number of training datasets, DPOT-S does not match the performance of MoE-POT-S. 2) For a fixed-capacity dense model, simply increasing the number of training datasets does not guarantee better performance. In fact, performance degraded on four of the six datasets when scaling from 6 to 12. This reinforces our central claim that mixing disparate PDE datasets in a dense model leads to performance degradation and highlights the rationale for our MoE architecture. We will add a more detailed analysis and these experiments to the final version.
>
> ---
>
> #### **Thanks again**
>
> We sincerely thank you again for your constructive and detailed feedback, which has helped us identify clear pathways to improve our paper. Should you have any further questions or require additional discussion, please don't hesitate to reach out. If we have adequately addressed your concerns, we would be grateful for your consideration in adjusting your evaluation score accordingly.

---

> ### Author Response · Authors · 2025-08-03
> **Updated Experimental Analysis on Handling Diverse PDE Datasets (Question 2)**
>
> To provide more definitive empirical evidence for our claims and to comprehensively address your question, we conducted a rigorous set of experiments comparing MoE-POT-S and the dense DPOT-S model as the heterogeneity of the pre-training data was increased. Due to the time constraints of the rebuttal period, these experiments were not yet complete at the time of our initial response. We are now pleased to share the finalized results. Our conclusions remain unchanged and are, in fact, strengthened by this new data.
>
> **Experimental Setup**
>
> For full transparency and fairness, we selected all datasets from the original DPOT paper's pre-training and downstream task sets. The specific mixtures were as follows:
>
> - 6 Datasets (This paper's pre-training set):
>   - NS(1e-5); 2. NS(1e-3); 3. CNS(0.1, 0.01); 4. SWE; 5. DR; 6. CFDBench
> - 10 Datasets (Added 4 diverse tasks):
>   - The 6 datasets above, plus: 7. NS(1e-4); 8. CNS(1, 0.1); 9. NavierStokes-2D (from PDEArena); 10. NavierStokes-2D-conditioned (from PDEArena)
> - 12 Datasets (Added 2 more diverse tasks):
>   - The 10 datasets above, plus: 11. CNS(1, 0.01); 12. CNS(0.1, 0.1)
>
> Crucially, for each configuration (6, 10, and 12 datasets), both DPOT-S and MoE-POT-S were **pre-trained entirely from scratch** on the specified data mixture; this was not a continuation of a previous training run.
>
> **Complete Experimental Results (L2RE)**
>
> Below is the updated table, which includes data from the original DPOT paper for reference. We have corrected a minor data misalignment from our previous response—please consider this version definitive.
>
> |                                 | NS(1e-5) | NS(1e-3) | CNS(0.1,0.01) | SWE    | DR     | CFDBench |
> | ------------------------------- | -------- | -------- | ------------- | ------ | ------ | -------- |
> | DPOT-S (6 datasets)             | 0.0688   | 0.0078   | 0.0244        | 0.0039 | 0.0367 | 0.0087   |
> | DPOT-S (10 datasets)            | 0.0663   | 0.0069   | 0.0224        | 0.0037 | 0.0575 | 0.0146   |
> | DPOT-S (12 datasets)            | 0.0739   | 0.0079   | 0.0129        | 0.0105 | 0.0724 | 0.0075   |
> | DPOT-S (12 datasets) (original) | 0.0553   | 0.0131   | 0.0188        | 0.0065 | 0.0379 | 0.0070    |
> | MoE-POT-S (6 datasets)          | 0.0552   | 0.0058   | 0.0096        | 0.0029 | 0.0342 | 0.0045   |
> | MoE-POT-S (10 datasets)         | 0.0534   | 0.0052   | 0.0085        | 0.0029 | 0.0371 | 0.0047   |
> | MoE-POT-S (12 datasets)         | 0.0563   | 0.0053   | 0.0062        | 0.0032 | 0.0383 | 0.0043   |
>
>
>
> **Analysis**
>
> These comprehensive results lead to two clear conclusions:
>
> 1. **Superior Performance of MoE-POT:** Regardless of the number of training datasets, MoE-POT-S consistently outperforms DPOT-S on a majority of the evaluation tasks.
> 2. **Robustness to** **Heterogeneity****:** For a fixed-capacity dense model like DPOT-S, simply increasing the number and diversity of training datasets does not guarantee better performance. Notably, as the dataset mixture grows more heterogeneous (from 6 to 12), the performance of **DPOT-S degrades significantly on the SWE and DR datasets**. In stark contrast, **MoE-POT-S** **maintains** **highly stable**.
>
> This experiment robustly reinforces our central claim: mixing disparate PDE datasets in a dense model leads to performance degradation due to "knowledge conflict," and it powerfully highlights the rationale for our sparse MoE architecture, which is specifically designed to overcome this challenge. We will add a more detailed analysis and these complete experimental results to the final version of the paper.

---

> > ### Comment · Reviewer_J1rL · 2025-08-06
> >
> > Thank you for the additional experiments. The rebuttal and fruitful discussion with other reviewers have convinced me that MoE-POT does provide a small edge over DPOT. This is also the first work on MoE for PDE surrogate learning tasks. Based on these positives, I raise my score to acceptance.

---

> > > ### Author Response · Authors · 2025-08-06
> > > **Thanks for Your Feedback and Revision Plan**
> > >
> > > We greatly appreciate your constructive feedback and your willingness to raise your scores! We are pleased to hear that our responses have resolved most of your concerns. As suggested, we will incorporate the newly added results into the final version of the paper.

---

### Decision · Program_Chairs · 2025-09-17

**Decision:**

Accept (poster)

**Comment:**

This manuscript proposes a new architecture that combines the concept of MoE with neural operators. It is to address the challenge of building foundational models capable of fitting multiple PDEs simultaneously. Whereas a naive approach would substantially increase computation, this work employs two types of expert networks to mitigate that overhead. Empirical evaluation demonstrates that the proposed architecture outperforms baselines and operates as expected.

The AC recommends: Accept as a poster.

**Strengths:** The main strength of the paper is being the first to employ the MoE concept within neural operators in the context of foundational models. All reviewers highlighted this point, and the AC concurs. There was extensive discussion during the review process regarding clarity and additional experiments, and the reviewers were ultimately satisfied with the rebuttals. Although a minor concern remains about the relatively modest performance improvement over prior work, the AC believes that the novelty lies in the architectural design and conceptual contribution rather than the raw performance gains.

**Final concern.** During the discussion, the authors provided substantial additional experimental results:

1. Comparison of computational overheads to DPOT
2. Performance analysis across different numbers of datasets
3. Model and activation parameter analysis
4. Stress-testing experiments

If incorporated directly into the main body of the camera-ready, these additions would go far beyond minor changes. The camera-ready version should not constitute a substantial revision. Therefore, the AC recommends that the authors restrict changes to minimal edits (e.g., typos, clarifications) in the main text, while placing additional experimental results in the Appendix.